# IRF3 prevents colorectal tumorigenesis via inhibiting the nuclear translocation of β-catenin

Miao Tian[1,7], Xiumei Wang[1,7], Jihong Sun[2,7], Wenlong Lin[1], Lumin Chen[2], Shengduo Liu[3], Ximei Wu [4], Liyun Shi[5], Pinglong Xu [3], Xiujun Cai [6✉] & Xiaojian Wang [1✉]

Occurrence of Colorectal cancer (CRC) is relevant with gut microbiota. However, role of IRF3, a key signaling mediator in innate immune sensing, has been barely investigated in CRC. Here, we unexpectedly found that the IRF3 deficient mice are hyper-susceptible to the development of intestinal tumor in AOM/DSS and Apc[min/+] models. Genetic ablation of IRF3 profoundly promotes the proliferation of intestinal epithelial cells via aberrantly activating Wnt signaling. Mechanically, IRF3 in resting state robustly associates with the active β-catenin in the cytoplasm, thus preventing its nuclear translocation and cell proliferation, which can be relieved upon microbe-induced activation of IRF3. In accordance, the survival of CRC is clinically correlated with the expression level of IRF3. Therefore, our study identifies IRF3 as a negative regulator of the Wnt/β-catenin pathway and a potential prognosis marker for Wnt-related tumorigenesis, and describes an intriguing link between gut microbiota and CRC via the IRF3-β-catenin axis.

[1] Institute of Immunology and Bone Marrow Transplantation Center, The First Affiliated Hospital, School of Medicine, Zhejiang University, 310003 Hangzhou, China. [2] Department of Radiology, Sir Run Run Shaw Hospital, School of Medicine, Zhejiang University, 310016 Hangzhou, China. [3] The MOE Key Laboratory of Biosystems Homeostasis & Protection and Innovation Center for Cell Signaling Network, Life Sciences Institute, Zhejiang University, 310058 Hangzhou, China. [4] Department of Pharmacology, School of Medicine, Zhejiang University, 310058 Hangzhou, Zhejiang, China. [5] Department of Immunology and Medical Microbiology, Nanjing University of Chinese Medicine, 210046 Nanjing, China. [6] Department of General Surgery, Innovation Center for Minimally Invasive Techniques and Devices, Sir Run Run Shaw Hospital, Zhejiang University School of Medicine, 310016 Hangzhou, China. [7] These authors contributed equally: Miao Tian, Xiumei Wang, Jihong Sun. ✉email: srrsh_cxj@zju.edu.cn; wangxiaojian@cad.zju.edu.cn

Colorectal cancer (CRC) is the third leading cause of cancer deaths worldwide. Causations for tumorigenesis and progression of CRC are complicated, and may include complex interactions among environmental exposures, diet, and heredity[1]. Many genetic and epigenetic alterations of proliferative signaling pathways and tumor suppressors are also characterized in the pathogenesis of CRC, such as the Wnt pathway, the TGF-β pathway, the (PI3K)-AKT pathway, the MAPK pathway, and the tumor protein p53 (TP53)[2]. In the case of Wnt signaling, β-catenin accumulates and translocates into the nucleus upon Wnt activation, where it binds TCF/LEF transcription factor and promotes the proliferation of intestinal stem cells that cause tumorigenesis[3].

Commensal microbes comprised of bacteria, archaea, viruses, and eukaryotes inhabit at all mucosal surfaces of the colon, which provide the physical barrier in defense against invading pathogens and modulate the gut environments[4]. Intriguingly, CRC tumorigenesis is frequently associated with the dramatic alteration in the microbial composition of the tumor and adjacent mucosa, commonly termed as dysbiosis. The emerging evidence reveals a critical role of *Fusobacterium nucleatum*[5], *Escherichia coli*[6,7], and *Bacteroides fragiles*[8] in colon tumorigenesis. However, the precise mechanism of gut microbiota in the initiation and progression of CRC are still largely unknown.

Intestinal microbiota induce innate immune responses through triggering of microbial sensors, namely the pathogen recognition receptors (PRRs)[9], including Toll-like receptors (TLRs), RIG-I-like receptors, NOD-like receptors (NLRs), C-type lectin receptors, and cytosolic DNA sensors. The adaptor proteins of these receptors activate the downstream protein kinases TBK1 and IKKs, which subsequently activates the transcription factor IRF3 and NF-κB, resulting in the production of type I IFNs and pro-inflammatory factors[10]. Notably, PRRs such as cGAS, TLRs, and NLRs[11], and the adaptors STING[12] and Myd88[13] are known to play crucial roles in maintaining the intestinal homeostasis and regulating the development of CRC, supposedly via their functions in secretion of type I IFNs, inflammatory cytokines, chemokines, and antimicrobial peptides[14]. However, the presence of alternative mechanism(s) of these innate immune elements in CRC tumorigenesis, such as independence of intestinal inflammation, is currently unknown.

IRF3 functions as both the signaling mediator and the transcription factor downstream several pathways of PRRs, and plays a key role in the production of type I and type III interferons, and a variety of IFN-stimulated genes (ISGs)[15]. IRF3 is ubiquitously expressed in cells with distinct origins, and resides in the cytoplasm in resting state that designed as an inactive form. Upon sensing the pathogen by PRRs, IRF3 is activated via carboxyl terminal phosphorylation by TBK1 and/or IKKε, which mobilizes IRF3 for dimerization and nuclear translocation, where it functions as the transcription factor[16]. Intriguingly, DNA damage promotes antitumor immunity via activating cGAS-STING-IRF3 pathway in cancer[17,18]. Hideo et al. found the activation of IRF3 by nucleic acid-sensing innate receptors is critical for intestinal homeostasis through its induction of protective Tslp and Il33 gene expression[19]. Previous report also indicates that activated IRF3 can attenuate TGF-β/Smad signaling, thus preventing in vivo differentiation of iTreg in colons and epithelial-to-mesenchymal transition of tumor cells, independent of its potency as transcription factor[20]. These intriguing observations implicate a close involvement of IRF3 in tumorigenesis of CRC, and it is worthy to investigate whether other important mechanism(s) exist.

Here, we found an intriguing function of cytosolic IRF3 in resting state to inhibit colorectal tumorigenesis via the prevention of Wnt/β-catenin signaling. IRF3 binds to the armadillo repeats (ARM), a domain crucial for β-catenin nucleus translocation, thus inhibiting the nuclear import of β-catenin. In accordance, IRF3 negatively correlates with the hyperactivation of Wnt signaling in tissues from CRC, lung adenocarcinoma, and hepatocellular carcinoma patients. Therefore, our data identify IRF3 as a tumor suppressor and a prognosis marker of the CRC patients with an unexpected mechanism.

## Results

**IRF3 in intestinal epithelium protects against colonic tumorigenesis.** IRF3$^{-/-}$ mice or wild-type littermates were administrated with azoxymethane (AOM) and dextran sulfate sodium (DSS). Substantially, more tumors and markedly increased tumor loads in colons of IRF3$^{-/-}$ mice were observed (Fig. 1a–d). In vivo magnetic resonance images (MRI) analyses also revealed a significantly increase of colon distension of IRF3$^{-/-}$ mice in both axial and coronal images, and more tumors in colons from IRF3$^{-/-}$ mice at day 90 (Supplementary Fig. S1a). Apc$^{min/+}$IRF3$^{-/-}$ mice displayed both more tumors and increased tumor load in the whole small intestine (Fig. 1e–h). In our Apc$^{min/+}$ mice model, the tumor were mainly located in the proximal small bowel (SB1; segments 1 and 2) and the distal small bowel (SB3; segments 2 and 3). The tumor located in the whole small intestine were summed. These data suggest that IRF3 protects mice against intestinal tumorigenesis.

Consistent with the previous report that deletion of epithelial Ifnar1 signaling in colon increases colitis-associated tumorigenesis[21], these IFNAR1$^{-/-}$ mice generated more tumors and increased tumor load (Supplementary Fig. S1b–e). Surprisingly, we observed that IRF3$^{-/-}$IFNAR1$^{-/-}$ mice were still more susceptible to tumorigenesis than IRF3$^{+/+}$IFNAR1$^{-/-}$ mice (Supplementary Fig. S1b–e), suggesting an additional function of IRF3 in CRC beyond of IFNAR signaling.

We next performed bone marrow chimaera studies in AOM/DSS model. Both the IRF3$^{-/-}$ → IRF3$^{+/+}$ mice and IRF3$^{+/+}$ → IRF3$^{-/-}$ mice, particularly the IRF3$^{+/+}$ → IRF3$^{-/-}$ mice, had significantly increased number and load of tumors compared to IRF3$^{+/+}$ → IRF3$^{+/+}$ mice (Fig. 1i–l). Consistently, in vivo MRI analyses revealed that IRF3$^{+/+}$ → IRF3$^{-/-}$ mice had significantly increased tumor burden than IRF3$^{-/-}$ → IRF3$^{+/+}$ mice (Supplementary Fig. S1f). These data suggest that the inhibitory effect of IRF3 on CRC is mostly determined by non-hematopoietic cells. Accordingly, IRF3$^{fl/fl}$Villin$^{cre}$ mice with conditional IRF3-knockout in intestinal epithelial cells developed the markedly higher number of tumors than control IRF3$^{fl/fl}$ mice (Fig. 1m–p). Taken together, these data suggest that the inhibitory effect of IRF3 on CRC is mainly through its expression in intestinal epithelial cells.

**Deficiency of IRF3 promotes proliferation of intestinal epithelial cells.** Paneth cells and goblet cells serve essential and specified functions to maintain the integrity of intestinal and colonic epithelium, as well as the stem cell population[22]. Thus, constant supply of both types of cell in appropriate ratios is critical for the homeostasis of intestinal system. To evaluate the role of IRF3 in the differentiation or proliferation of Paneth cells or goblet cells in the small intestinal and colon, we stained lysozyme-IHC or Alcian blue/periodic acid Schiff to measure their number in IRF3$^{-/-}$ mice. However, no significant difference was found in numbers of Paneth or Goblet cells in small intestine, and colon between wild-type and IRF3$^{-/-}$ mice (Supplementary Fig. S2a, b). Development and progression of CRC are regulated by the composition of gut microbiota[23]. However, no difference was found in community diversity (Supplementary Fig. S2c) and structure (Supplementary Fig. S2d) of the Decal microbiota in IRF3$^{+/+}$ and IRF3$^{-/-}$ mice, as evidenced by the 16 S rRNA sequencing results. Co-housed experiment also showed that the co-housed IRF3$^{-/-}$ mice were still more susceptible to CRC upon AOM/DSS treatment (Supplementary Fig. S2e–h). As revealed in Supplementary Fig. S2i, deficiency of IRF3 had no significant effect on the expression of

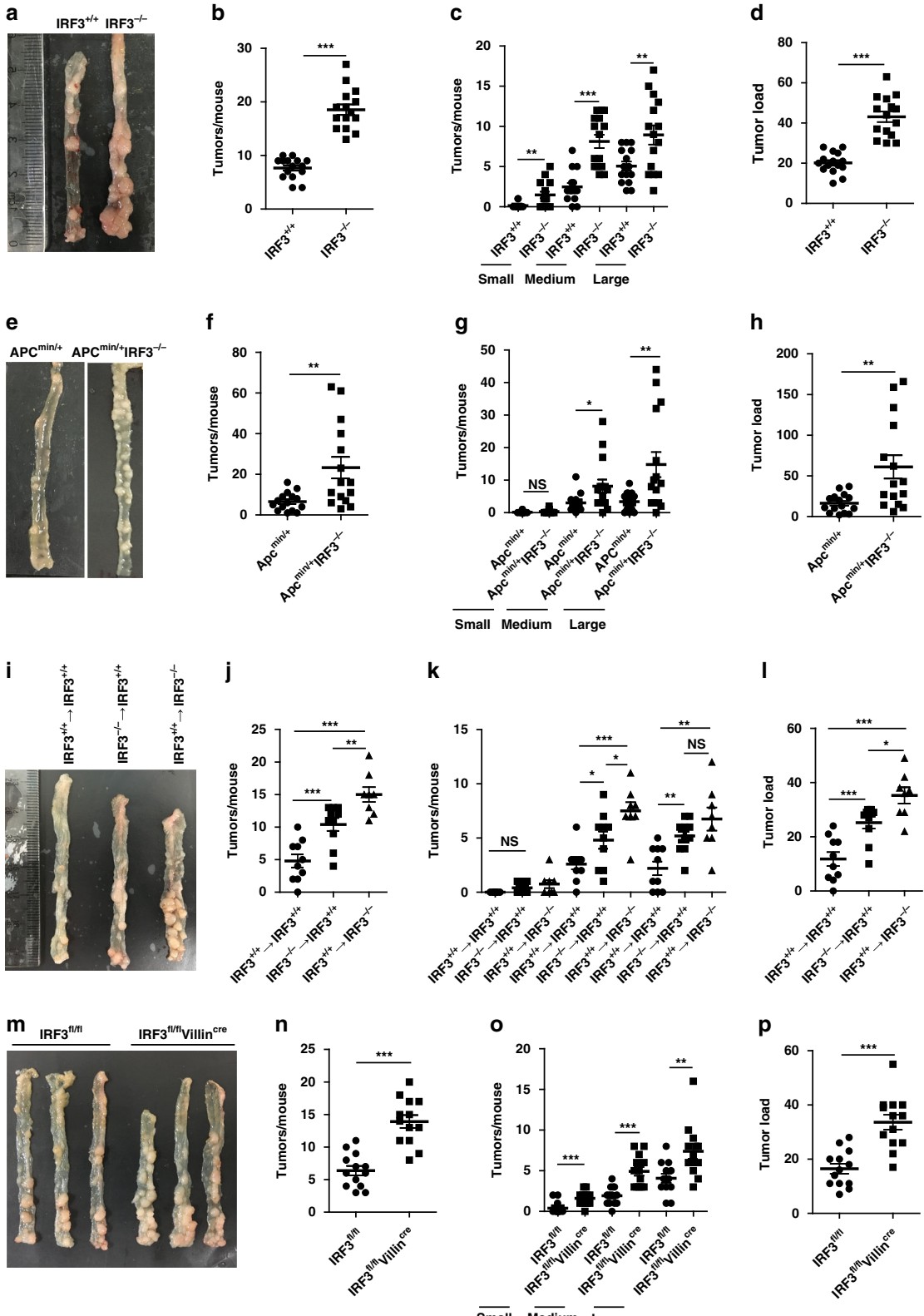

**Fig. 1 IRF3 in intestinal epithelium protects against colonic tumorigenesis. a** Representative images of colon tumors from IRF3$^{+/+}$ and IRF3$^{-/-}$ mice on day 90 after AOM/DSS model. **b–d** Colon tumor counts, size, and tumor load from IRF3$^{+/+}$ and IRF3$^{-/-}$ mice ($n = 15$ mice/group) after AOM/DSS model (day 90). **e** Representative images of the small intestine and tumors in Apc$^{min/+}$ and Apc$^{min/+}$IRF3$^{-/-}$ mice. **f–h** Intestinal tumors counts, size, and tumor load from Apc$^{min/+}$ and Apc$^{min/+}$IRF3$^{-/-}$ mice ($n = 15$ mice/group). **i–l** Three groups of mice were generated by bone marrow transplantation: IRF3$^{+/+}$ → IRF3$^{+/+}$, $n = 10$; IRF3$^{-/-}$ → IRF3$^{+/+}$, $n = 10$; IRF3$^{+/+}$ → IRF3$^{-/-}$, $n = 8$; the numbers and size of tumors in the colon were quantified after AOM/DSS model (day 90). **m–p** Colon tumor counts, size, and tumor load from IRF3$^{fl/fl}$ and IRF3$^{fl/fl}$ Villin$^{cre}$ mice ($n = 13$ mice/group) representative images of colons at left (**m**) after AOM/DSS model (day 90). Each symbol represents one mouse (**b–d**, **f–h**, **j–l**, **n–p**). *$P < 0.05$; **$P < 0.01$; ***$P < 0.001$; NS not statistically significant by two-tailed $t$ test (**b–d**, **f–h**, **j–l**, **n–p**). Data are from two independent experiments (**a–p**) and are presented as mean ± s.e.m. in **b–d**, **f–h**, **j–l**, **n–p**. See also Supplementary Fig. S1.

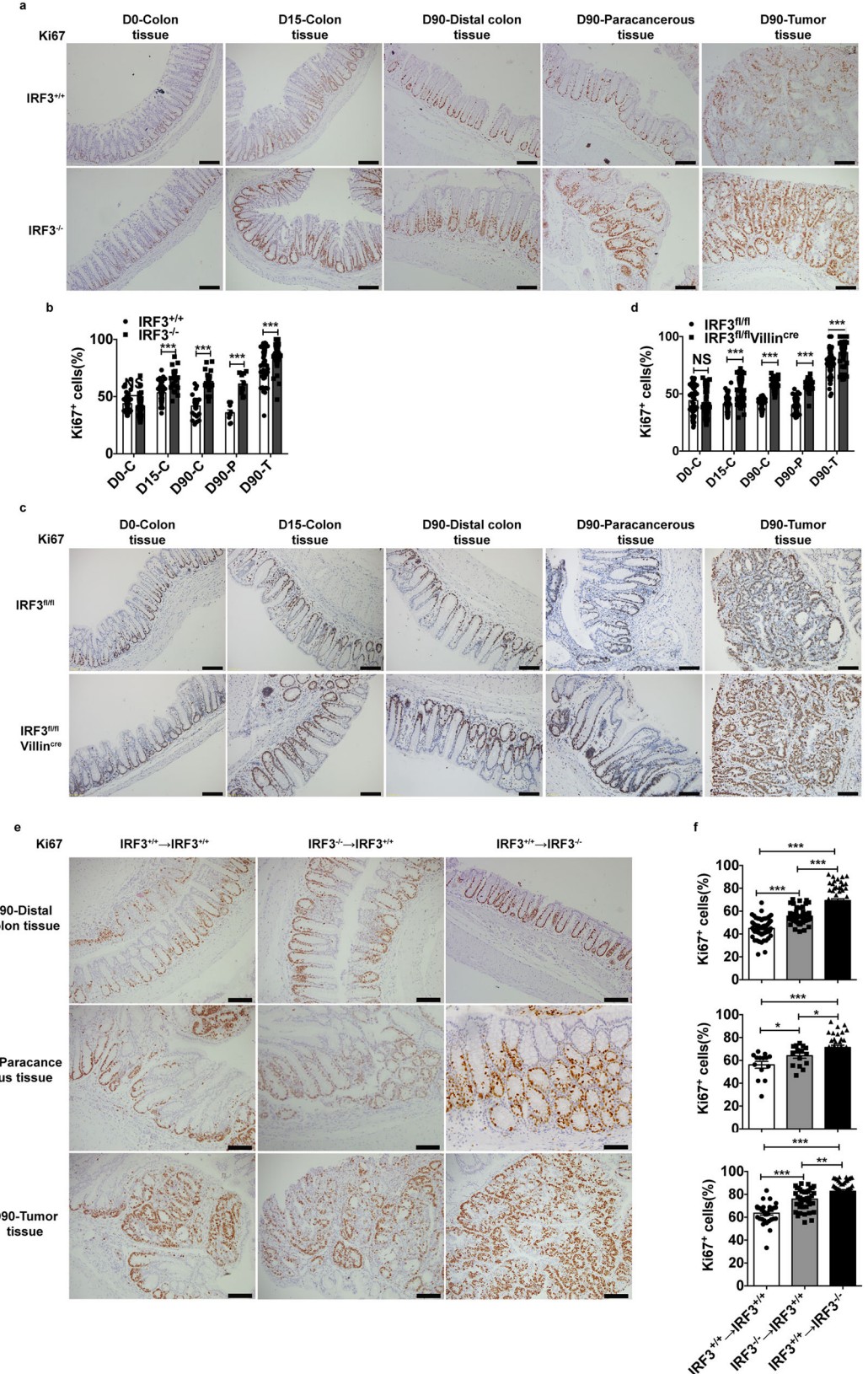

cytokines IL6, TNFα, and IL1β, and chemokines CXCL1 and CCL20 in the distal colon and tumor tissues. These observations suggest that CRC tumorigenesis due to the IRF3 ablation may not due to gut microbiota or the inflammation.

We then examined both apoptosis and proliferation of colon cells in IRF3$^{-/-}$ or IRF3$^{fl/fl}$Villin$^{cre}$ mice. Genetic ablation of IRF3 had no

effect on the apoptosis of enterocytes, as detected by TUNEL assay (Supplementary Fig. S2j) and cleaved-caspase 3 staining (Supplementary Fig. S2k). However, we detected a higher level of ki67$^+$ cells per crypt in the distal colon, para-cancerous and tumor tissues from IRF3$^{-/-}$ mice and IRF3$^{fl/fl}$Villin$^{cre}$ upon AOM treatment (Fig. 2a–d), or from the bone marrow chimaera mice IRF3$^{+/+}$ →

**Fig. 2 Deficiency of IRF3 promotes the proliferation of intestinal epithelial cells. a** Standardized Ki67 immunostaining of the distal colon, and tumors from IRF3[+/+] and IRF3[−/−] mice on day 0, 15, and 90 after AOM injection. Scale bar, 100 μm. **b** Quantification of the number of Ki67[+] in each crypt from IRF3[+/+] and IRF3[−/−] mice (day 0, $n = 3$; day 15, $n = 5$; day 90, $n = 5$). **c** Standardized Ki67 immunostaining of the distal colon and tumors from IRF3[fl/fl] and IRF3[fl/fl]Villin[cre] mice on day 0, 15, and 90 after AOM injection. Scale bar, 100 μm. **d** Quantification of the number of Ki67[+] in each crypt from IRF3[fl/fl] and IRF3[fl/fl]Villin[cre] mice (day 0, $n = 3$; day 15, $n = 4$; day 90, $n = 5$). **e** Standardized Ki67 immunostaining of the distal colon and tumors from chimera on day 90 after AOM injection. Scale bar, 100 μm. **f** Quantification of the number of Ki67[+] in each crypt of chimera mice (day 90, $n = 3$ mice/group). *$P <$ 0.05; **$P < 0.01$; ***$P < 0.001$; NS not statistically significant by two-tailed $t$ test (**a–f**). Data represent two independent experiments (**a–f**) and are presented as mean ± s.e.m. in **a–f**. See also Supplementary Fig. S2.

IRF3[−/−] (Fig. 2e–f). These observations were consistent with the observed higher tumor load in these mice (Fig. 1i–l). In addition, we evaluated the xenograft growth of tumor cells in nude mice transplanted from the AOM/DSS-treated mice, to further exclude the involvement of inflammation effect. Tumor harboring IRF3[−/−] genetic modification grew faster and gained marked weight after transplantation (Supplementary Fig. S2l–n). These observations suggest that deficiency of IRF3 promotes proliferation of intestinal epithelial cells.

**IRF3 suppresses the CRC via inhibiting Wnt signaling.** We employed an RNA-sequencing approach to interrogate the pathways differentially regulated in distal colon and tumor tissues, upon AOM treatment and in the presence or absence of IRF3. We obtained 92 genes upregulated in tumor tissue compared to distal colon both in "WT" and "KO" groups, 16 genes only elevated in "WT" group, and 65 genes upregulated in "KO" group (Supplementary Fig. S3a). Alternation of these 65 genes revealed that deficiency of IRF3 markedly changed the Wnt pathway in colon tumorigenesis, by analyzing with PANTHER database (http://www.pantherdb.org/; Fig. 3a).

We then examined the nucleus translocation of β-catenin in CRC, a proof for activation of Wnt signaling. As shown in Fig. 3b, the level of nuclear β-catenin in IRF3[−/−] cells was higher than wild-type mice upon AOM treatment. Quantification of mRNA revealed that target genes of Wnt signaling, including c-Myc, Cyclin D1, Axin2, Lef1, and TCF1, as well as Wnt-associated stem cells markers, including Lgr5, Ascl2, and CD44v6, were higher expressed in IRF3[fl/fl]Villin[cre] mice in AOM/DSS model (Fig. 3c, d and Supplementary Fig. S3b, c). Similar increasement was also observed in IRF3[−/−] mice upon AOM treatment (Supplementary Fig. S3d–g). These results showed that IRF3 difficiency did not affect the basal level of Wnt target or associated genes expression in the day 0 intestine tissue. Meanwhile, we performed RNA in situ hybridization for stem cell marker (Lgr5) and Wnt target gene (Axin2) in normal crypts (SI and colon) from IRF3[fl/fl] and IRF3[fl/fl]Villin[cre] mice. As shown in the Supplementary Fig. S3h, the RNA level of Lgr5 and Axin2 in normal crypts showed no difference in IRF3[fl/fl] and IRF3[fl/fl]Villin[cre] mice. Isolated primary colonic stem cells can develop into sphere-like "organoids", which relies on the hyperactive Wnt signaling[24]. As shown in Fig. 3e, colonic epithelial stem cells collected from the IRF3[−/−] mice were more readily developed into organoids in vitro, with higher organoid numbers and enlarged diameters (Fig. 3f). We simultaneously analyzed the phosphorylation/activated state of STAT3, Akt, Erk1/2, and p38 in the distal colon and tumor tissues from IRF3[+/+], IRF3[−/−], IRF3[fl/fl], and IRF3[fl/fl]Villin[cre] mice following AOM/DSS treatment. As shown in Supplementary Fig. S3i, j, IRF3 deficiency did not affect levels of phosphorylated forms of these proteins. A recent report indicated that IRF3 interacted with both Yes-associated proteins (YAP) and TEAD4 in the nucleus and promoted YAP activation, resulting in accelerating gastrointestinal carcinoma progression[25], but we failed to detect such change of YAP in the absence of IRF3, either by cellular distribution of YAP (Supplementary Fig. S3k) or by

mRNA expression level of YAP/TAZ target genes (CTCG and Cyr61; Supplementary Fig. S3l).

Furthermore, treatment of specific Wnt inhibitor ICG-001[26] and G007-LK[27] abolished the increase of tumorigenesis (Fig. 3g–k and Supplementary Fig. S3n–p) and Wnt signaling (Supplementary Fig. S3m, q) in IRF3[fl/fl]Villin[cre] mice. Collectively, these data suggest that IRF3 limits colon tumorigenesis via suppressing the Wnt/β-catenin pathway.

**Cytoplasmic IRF3 in resting state inhibits Wnt signaling and epithelial cell proliferation.** IRF3-knockout cells were generated in human colon cell line HCT116 and SW620, as well as human non-small lung carcinoma cell line H1299, as Wnt signaling is also critical in non-small cell lung cancer[28]. IRF3-defeciency HCT116, H1299, and SW620 cells exhibited the clearly increased levels of cell proliferation (Fig. 4a, b and Supplementary Fig. S4a) and colony formation (Fig. 4c and Supplementary Fig. S4b) compared to parent cells. qPCR analyses revealed higher Wnt signaling activation in IRF3-knockout cells upon serum stimulation (Supplementary Fig. S4c–f). Downregulation of IRF3 in HCT116, SW620, and H1299 cells reached similar phenotypes for cell proliferation (Supplementary Fig. S4g) and mRNA expression (Supplementary Fig. S4h–k). These data suggest that deficiency or downregulation of IRF3 results in the enhanced Wnt signaling and cell proliferation.

Consistently, ICG-001 treatment downregulated the proliferation of IRF3-knockout cells to a level similar with wild-type cells (Fig. 4d), and tumor formation assays phenocopied this observation (Fig. 4e, f), with marked decrease of Ki67-positive cells (Fig. 4g) and reduced Wnt signaling activation (Supplementary Fig. S4l). Enhanced tumor formation of IRF3-knockout H1299 cells was also abolished by ICG-001 treatment (Supplementary Fig. S4m, n), as well as Wnt signaling activation (Supplementary Fig. S4o). Importantly, depletion of IRF3 failed to affect the proliferation of HCT116 cells with β-catenin-knockout (Ctnnb1[−/−]; Fig. 4h and Supplementary Fig. S4p). These consistent observations suggest that Wnt/β-catenin underlies the IRF3-mediated suppression of tumor cell growth.

IRF3 shuttles between the cytoplasm and the nucleus depending on its C-terminal phosphorylation by TBK1/IKKε. We generated IRF3 mutants including IRF3-ΔnDB that lacks the DNA-binding domain[20], IRF3-ΔNLS that lacks the nuclear localization signal (NLS)[29] and the IRF3-5D that mimics constitutive activate IRF3[30]. Ectopic expression of IRF3, IRF3-ΔnDB, or IRF3-ΔNLS, but not IRF3-5D mutant, significantly decreased the cell proliferation and activation of Wnt signaling in HCT116, SW620, and H1299 cell lines (Supplementary Fig. S4q–t). Consistently, reintroduction of IRF3, IRF3-ΔnDB, or IRF3-ΔNLS, but not IRF3-5D, restored the cell proliferation (Fig. 4i, j) and Wnt signaling activation (Fig. 4k and Supplementary Fig. S4u) in IRF3-knockout cells. Tumor formation assays in nude mice with IRF3-knockout HCT116 cells, with or without IRF3 rescue, revealed that reintroduction of IRF3, IRF3-ΔnDB, or IRF3-ΔNLS, but not the IRF3-5D, exhibited faster growth (Fig. 4l, m). These data suggest that IRF3 capability to

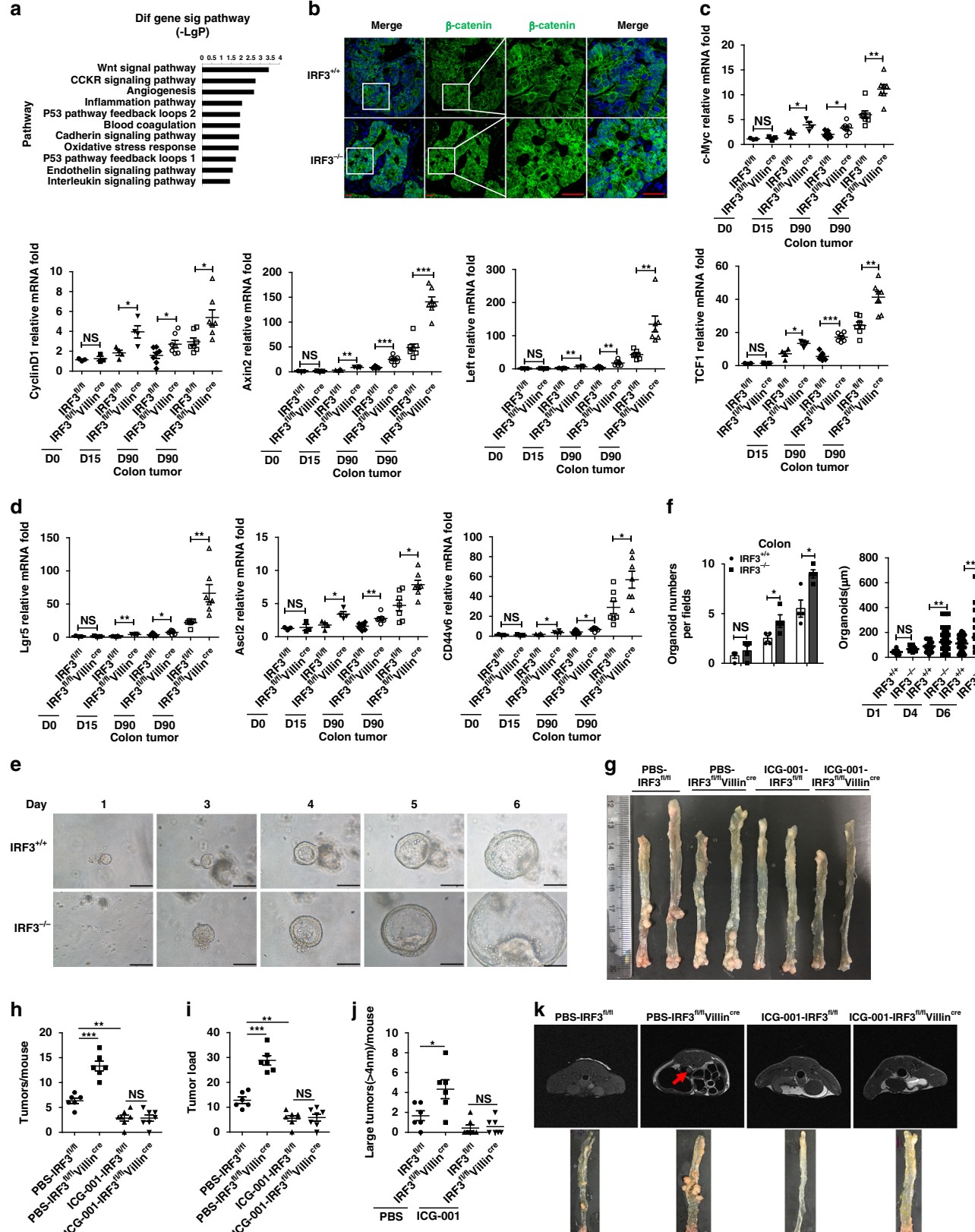

inhibit Wnt signaling and cell proliferation requires its resting state.

Luciferase reporter assays also showed that co-transfection of β-catenin-S33A mutant[31], with wild-type IRF3, IRF3-ΔnDB, or IRF3-ΔNLS, but not IRF3-5D, inhibited transactivation of β-catenin (Supplementary Fig. S4v). Intriguingly, activation of IRF3

by constitutive active RIG-I (RIG-I-N; Supplementary Fig. S4w) rendered both IRF3 and IRF3-ΔnDB to lose their capabilities to inhibit β-catenin signaling (Supplementary Fig. S4v). Similar observations were also retrieved in HCT116 cells (Supplementary Fig. S4x). These observations thus suggest that the cytosolic IRF3 in resting state suppresses β-catenin transactivation.

**Fig. 3 IRF3 suppresses the CRC via inhibiting Wnt signaling. a** The signal pathways were enriched with the 65 genes that upregulated in tumor tissue only in "KO" from the RNA-seq analysis results. **b** Immunofluorescence analysis of β-catenin nuclear translocation in colorectal tumors from IRF3$^{+/+}$ and IRF3$^{-/-}$ mice after treatment with AOM/DSS (days 0 and 90). Scale bar, 20 μm. **c, d** Real time qPCR analysis for expression of the Wnt target, and associated genes in the distal colon and tumors from IRF3$^{fl/fl}$ and IRF3$^{fl/fl}$Villin$^{cre}$ (day 0, $n = 3$ mice/group; day 15, $n = 4$ mice/group; day 90, $n = 7$ mice/group) mice. **e–f** Images (**e**) and quantifications (**f**) of the number (left) and size (right) of organoids from IRF3$^{+/+}$ and IRF3$^{-/-}$ colon stem cells. **g** Representative images of colon tumors from IRF3$^{fl/fl}$ and IRF3$^{fl/fl}$Villin$^{cre}$ mice on day 90 after AOM/DSS model with pbs or ICG-001 treatment. **h–j** Colon tumors counts, size, and tumor load in AOM/DSS-treated mice with PBS or ICG-001 treatment (300 mg/kg per day, orally, once daily, six times 1 week for the last 10 weeks of the AOM/DSS model; PBS group, $n = 6$ mice/group; ICG-001 group, $n = 7$ mice/group). **k** Representative MRI images of IRF3$^{fl/fl}$ and IRF3$^{fl/fl}$Villin$^{cre}$ mice with PBS or ICG-001 treatment (300 mg/kg per day, orally, once daily, six times 1 week for the last 10 weeks of the AOM/DSS model). Arrowhead indicates colon tumor. Each symbol represents one organoid (**e**) or an individual mouse (**c, d, h–j**). *$P < 0.05$; **$P < 0.01$; ***$P < 0.001$; NS not statistically significant by two-tailed $t$ test (**c–f, h–j**). Data represent two (**b–d, g–k**) or three independent experiments (**e, f**), and are presented as mean ± s.e.m. in **a–j**. See also Supplementary Fig. S3.

**IRF3 binds with and prevents β-catenin nucleus translocation.** In resting state, GSK3β sequentially phosphorylates β-catenin that resulting in its ubiquitination and proteasomal degradation[24]. Binding of wnt3a to Frizzled receptor and LRP5/6 co-receptor leads to dephosphorylation of β-catenin that drives its nuclear translocation, where it interacts with TCF/LEF transcription factors[3]. Genetic ablation of IRF3 showed no effect on the protein level of the total β-catenin, the active (non-phospho)-β-catenin, and the phospho-β-catenin (33/37/41)[32], but promoted expression of Wnt target genes c-Myc and cyclin D1, upon wnt3a treatment in HCT116 cells (Supplementary Fig. S5a, b). Ectopic expression of IRF3 attenuated this induction of c-Myc and cyclin D1 (Supplementary Fig. S5c), while IRF3 deletion promoted the nuclear translocation of active-β-catenin (Fig. 5a, b) and that was reversed by overexpression of IRF3 (Supplementary Fig. S5d, e) in both HCT116 and H1299 cells. Immunofluorescent imaging also revealed the lesser nuclear β-catenin in cells overexpressing IRF3 upon wnt3a stimulation (Supplementary Fig. S5f). Notably, we observed that endogenous IRF3 is physically associated with both β-catenin and active-β-catenin, but not with GSK3β, in HCT116 cells (Fig. 5c) or H1299 cells (Supplementary Fig. S5g). Increased IRF3 association with active-β-catenin appeared to be related with its reduced interaction with total β-catenin upon wnt3a stimulation (Fig. 5c and Supplementary Fig. S5g). These data suggest an association between IRF3 and active-β-catenin underlines IRF3-mediated suppression of Wnt signaling.

Consistently with previous reports[33], Flag-IRF3 is co-immunoprecipitated with HA-β-catenin, but not with HA-GSK3β in HEK293T cells (Supplementary Fig. S5h). Notably, IRF3 was higher affinity with the active form of β-catenin (S33A)[31] (Fig. 5d). GST pull-down assays also revealed that β-catenin directly interacted with IRF3, while the active mutant of β-catenin had stronger interaction (Supplementary Fig. S5i), suggesting that activation of β-catenin facilitates its interaction with IRF3.

Domain mapping revealed that C-terminus (a.a. 634–781) and ARM repeats (a.a. 133–695) of β-catenin, which are required for nucleus translocalization of β-catenin[34], were required for IRF3 interaction (Supplementary Fig. S5j, k). More accurately, ARM repeats itself was sufficient to interact with IRF3 (Fig. 5e and Supplementary Fig. S5k). The segment of a.a. 634–663 of β-catenin was further identified that binding to IRF3 (Fig. 5e and Supplementary Fig. S5l) and requiring for its nucleus translocation upon activation, as evidenced by immunofluorescent imaging or nucleocytoplasmic separation assay (Fig. 5f, g). Together, these data indicate that IRF3 binds with the segment of a.a. 634–663 of β-catenin, an interface required for nuclear translocation of β-catenin.

We further mapped that a.a. 357–427 segment of IRF3 was responsible for their interaction (Supplementary Fig. S5m). Thereby, rescue with IRF3-ΔC in IRF3-knockout cells failed to restore the enhanced cell proliferation and elevated Wnt signaling (Supplementary Fig. S5n–p), and ectopic expression of IRF3-ΔC failed to inhibit the transactivation of β-catenin-S33A (Supplementary Fig. S5q). These data suggest that the C-terminus motif (a.a. 357–427) of IRF3 is required for β-catenin interaction Wnt signaling suppression.

**Activation of IRF3 by PRR signaling relieves its inhibition on Wnt signaling.** Since mimicking IRF3 activation forfeited inhibitory effect of IRF3 on Wnt signaling and cell proliferation (Fig. 4i–k and Supplementary Fig. S4q–u), we speculated that virus infection, which activates IRF3, may activate Wnt signaling. Vesicular stomatitis virus (VSV) infection activated the Wnt pathway in HCT116 cells, as evidenced by luciferase report assay, nucleocytoplasmic separation assay, and QPCR assay (Fig. 6a, b and Supplementary Fig. S6a). Intriguingly, Wnt activation by VSV infection was not affected by IRF3-knockout or knockdown (Fig. 6a–c and Supplementary Fig. S6a, b), but was greatly inhibited by the treatment using cycloheximide (new protein synthesis inhibitor), U0126 (MAPK signaling inhibitor), or BAY 11-7082 (NF-κB signaling inhibitor; Supplementary Fig. S6c). In line with the previous report[35], IRF3 deficiency significantly inhibited type I interferon and ISG expression, but had no effect on IL6 and TNFα expression (Supplementary Fig. S6d, e). Taken together, these results indicated that virus-mediated activation of Wnt signaling is probably due to two mechanisms: on one hand, virus-induced IRF3 activation relieves its inhibition on Wnt signaling; on the other hand, virus infection induces an unknown protein via MAPK and NF-κB signaling, which in turn mediates Wnt activation.

We then hypothesized that C-terminus phosphorylation of IRF3 influenced its interaction with β-catenin. Co-immunoprecipitation experiments showed that IRF3-5D failed to associate with HA-β-catenin in HEK293T cells (Fig. 6d), and similarly, HA-β-catenin-S33A was co-immunoprecipitated with Flag-IRF3, but not with Flag-IRF3-5D (Fig. 6e). Notably, endogenous IRF3 was physically associated with β-catenin and active-β-catenin in the HCT116 and H1299 cells, and this association was decreased following IRF3 activation triggered by VSV treatment (Fig. 6f and Supplementary Fig. S6f), further supporting that the resting state IRF3, but not its activated form, associates with and facilitate cytoplasmic retention of active-β-catenin.

Sensing of gut microbiota PAMPs by TLRs also activates signaling pathways to IRF3 activation[36]. Ongoing research has confirmed that gut microbiota links to CRC tumorigenesis[23] and antibiotics inhibits the development of intestinal tumor in Apc$^{min/+}$ mice[37]. We then applied a cocktail of antibiotics (Abx) in this spontaneous intestinal cancer model, which revealed that 4 months treatment of Abx decreased tumorigenesis in both Apc$^{min/+}$ mice and Apc$^{min/+}$IRF3$^{-/-}$ mice (Fig. 6g, h). However,

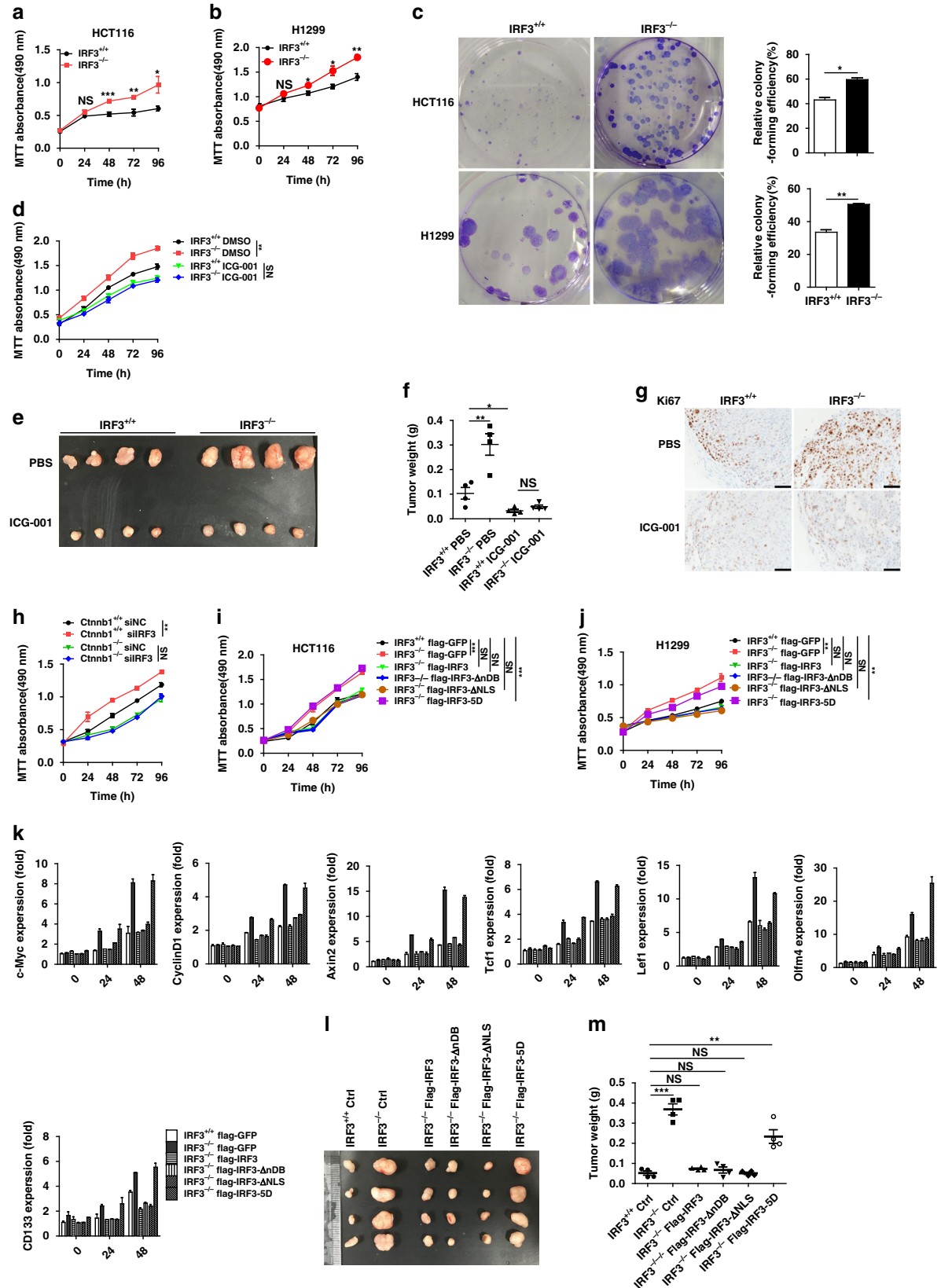

the decreased rate of tumor number in Apc$^{min/+}$ mice (67.4%) was higher than those in Apc$^{min/+}$ IRF3$^{-/-}$ mice (55%; Fig. 6h): Abx treatment reduced the average number of tumors per mice from 8.1 to 2.6 in Apc$^{min/+}$ mice, while reduced the number of

tumors per mice from 22.9 to 10.3 in the Apc$^{min/+}$IRF3$^{-/-}$ mice. We had performed two independent experiments that the Apc$^{min/+}$ and Apc$^{min/+}$IRF3$^{-/-}$ mice were treated with or without Abx. As shown in Supplementary Table S1, the reduction of polyps in

**Fig. 4 The cytoplasmic IRF3 in resting state inhibits the cell proliferation and Wnt/β-catenin pathway in HCT116 and H1299 cell lines. a, b** Proliferation of the IRF3$^{+/+}$ and IRF3$^{-/-}$ HCT116 (**a**) and H1299 (**b**) cells. **c** Colony formation experiment of the IRF3$^{+/+}$ and IRF3$^{-/-}$ HCT116 and H1299 cells. **d** Proliferation of the HCT116 cells with Wnt signaling inhibitor ICG-001 (50 μM) or DMSO treatment. **e, f** Representative images of tumors from subcutaneous tumor formation assay in nude mice (**e**). Subcutaneous tumor formation assay in nude mice with 2 × 10$^6$ IRF3$^{+/+}$ or IRF3$^{-/-}$ HCT116 cells per mouse. After 1 week of the injection, PBS or ICG-001 treatment (200 mg/kg, i.v., once daily) was applied in mice until the end of the model. Tumor weight for each group ($n = 4$) was plotted (**f**) at day 21 after injection. **g** Immunohistochemical analysis for ki67 in tumors from **e**. **h** Proliferation of β-catenin-wild type (Ctnnb1$^{+/+}$) and β-catenin-knockout (Ctnnb1$^{-/-}$) HCT116 cells treated with siNC or siIRF3. **i, j** Proliferation of the IRF3$^{+/+}$ and IRF3$^{-/-}$ HCT116 (**i**) and H1299 (**j**) cells transfected with the indicated plasmids expressing backbone, IRF3, IRF3-ΔnDB, IRF3-ΔNLS, or IRF3-5D. **k** Real time qPCR analysis for the Wnt target, and associated genes in IRF3$^{+/+}$ and IRF3$^{-/-}$ HCT116 cells transfected with the indicated plasmids expressing backbone, IRF3, IRF3-ΔnDB, IRF3-ΔNLS, or IRF3-5D. **l, m** The stable expression control plasmid, Flag-tagged IRF3, -IRF3-ΔnDB, -IRF3-ΔNLS, and IRF3-5D mutations IRF3$^{+/+}$ or IRF3$^{-/-}$ HCT116 cells were applied to subcutaneous tumor formation assay in nude mice with 2 × 10$^6$ cells/group per mouse for 21 days. Images of tumor grafts from these cells at day 21 (**l**). Tumor weight for each group ($n = 4$) was plotted in **m**. Each symbol represents one mouse (**m**). *$P < 0.05$; **$P < 0.01$; ***$P < 0.001$; NS not statistically significant by two-tailed $t$ test (**a–m**). Data represent two (**e–g**, **l**, **m**) or three independent experiments (**a–d**, **h–k**) and are presented as mean ± s.e.m. in **a–m**. See also Supplementary Fig. S4.

Apc$^{min/+}$ mice was more than that in Apc$^{min/+}$IRF3$^{-/-}$ mice with Abx treatment. This result indicates that triggered the activation of IRF3 by gut microbiota contributes to the microbiota-induced tumorigenesis. Simarly, the decreased rate of mRNA expression of Wnt target genes was higher in Apc$^{min/+}$ mice than those in Apc$^{min/+}$ IRF3$^{-/-}$ mice upon Abx treatment (Supplementary Fig. S6g). Meanwhile, Abx treatment inhibited the expression of type I interferon (IFNβ), and ISGs (MX1, ISG15, and IFIT1) in small intestine and tumor tissues, validating its regulation on IRF3 activation (Supplementary Fig. S6h). IHC assay also verified the reduced TCF1 and MX1 expression in Abx-treated Apc$^{min/+}$ mice (Fig. 6i and Supplementary Fig. S6i). Taken together, these observations suggest that IRF3 activation triggered by gut microbiota links to colon tumorigenesis.

**Expression level of IRF3 correlates with Wnt signaling activation and CRC patient survival.** To evaluate the correlation of IRF3 expression and Wnt signaling in human cancers, we examined protein expression levels of IRF3, TCF1[38], and LEF1[39] in human CRC ($n = 115$) and lung adenocarcinoma ($n = 67$) by tissue microarray-based IHC. Quantitatively standardized IHC analyses revealed that expression level of IRF3 protein was inversely correlated with levels of TCF1 and LEF1 (Fig. 7a, b and Supplementary Fig. S7a, b). Tumor tissues were then divided into groups with high and low levels of IRF3, TCF1, and LEF1 according to IHC scores. Notably, the low expression of IRF3 and the high expression of TCF1 and LEF1 were found to be significantly associated with the poor outcome in CRC (Fig. 7c) and lung adenocarcinoma patients (Fig. 7d). Furthermore, patients classified as IRF3$^{high}$/TCF1$^{low}$/LEF1$^{low}$ showed better disease outcome (Fig. 7e, f). Similar observations were retrieved from human hepatocellular carcinoma patients ($n = 92$), with inverse relation between IRF3 and TCF1/LEF1 (Fig. 7g and Supplementary Fig. S7c), and shorter survival time of patients with low IRF3 and high TCF1 and LEF1 expression (Fig. 7h). Consistently, the IRF3$^{high}$/TCF1$^{low}$/LEF1$^{low}$ group showed longer survival (Fig. 7i). These data indicate that higher protein level of IRF3 is associated with longer survival of CRC, lung adenocarcinoma, and hepatocellular carcinoma patients, probably via the negatively regulation of the Wnt pathway.

## Discussion
Here, we use an autochthonous mouse model that faithfully recapitulates molecular mechanism, pathology, and the progression of human CRC to determine the biological importance of the IRF3 in CRC development. We identify IRF3 as a tumor suppressor via inhibiting Wnt signaling, by an unexpected function that departs from its well-known role as transcription factor. We

demonstrated that IRF3 binds to the ARM domain of β-catenin to inhibit its nucleus translocation, thus resulting in the decreased expression of Wnt target genes and impeded cell proliferation. In accordance, deficiency of IRF3 promotes colorectal tumorigenesis by activating Wnt signaling, while CRC patients with lesser IRF3 expression correlate with enhanced Wnt signaling and poor survival.

Upon activation by Wnt ligands, the intrinsic kinase activity of the APC complex for β-catenin phosphorylation is inhibited, and stable and non-phosphorylated β-catenin is then accumulated and translocates into the nucleus, where it binds to the TCF/LEF transcription factors and drives transcription[3]. The link between levels of nuclear β-catenin and advancing stages of human colorectal carcinogenesis is well established, leading to the shorter survival of patients[40]. Therefore, targeting the nuclear β-catenin is an emerging anticancer strategy. The residues on β-catenin essential for its nuclear translocation have also been identified; residing in the 10th–12th ARM repeats[41]. Intriguingly, we found that the a.a. 634–663, in these exact ARM repeats of β-catenin, is necessary for binding IRF3, which linking IRF3-β-catenin interaction and cytoplasmic retention of β-catenin. Although the interaction between transfected IRF3 and β-catenin was reported[33], we demonstrated here the endogenous association of β-catenin with IRF3, particularly the active form of β-catenin. It was reported that an optimal but not excessive level of accumulation of nuclear β-catenin is considered favorable for tumorigenesis[42]. In our study, IRF3 deficiency promoted the tumorigenesis both in AOM/DSS and Apc$^{min/+}$ mouse model, indicating the IRF3-promoting β-catenin activation was in the optimal range of Wnt activation.

PRRs, the sentinel receptors for microbial invasion, are important in tumorigenesis. Our current findings suggest that the activation of PRR signaling in intestinal cells promote CRC tumorigenesis by relieving IRF3-inhibited Wnt signaling. Serial phosphorylation of IRF3 at C-terminus reorganizes auto-inhibitory elements of IRF3, leading to unmasking of a hydrophobic active site and realignment of the DNA-binding domain. The phosphorylated and acidic C-terminal tails of IRF3 are also stabilized in a dimer arrangement, through interactions with the basic surface of the neighboring IRF3 molecule[43,44]. We found there that the constitutively active IRF3 (IRF3-5D) fails to bind β-catenin or active-β-catenin (β-catenin-S33A), indicating that the structural/interface change of activated IRF3 abrogates its interaction with active-β-catenin, and thus loses its capacity to retain β-catenin in cytoplasm. Therefore, IRF3 activation triggered by gut microbiota, relieves the inhibitory effect of IRF3 on Wnt signaling, while Abx treatment substantially inhibits the spontaneous intestinal carcinogenesis. It thus links the gut environment with CRC tumorigenesis via the IRF3-β-catenin axis.

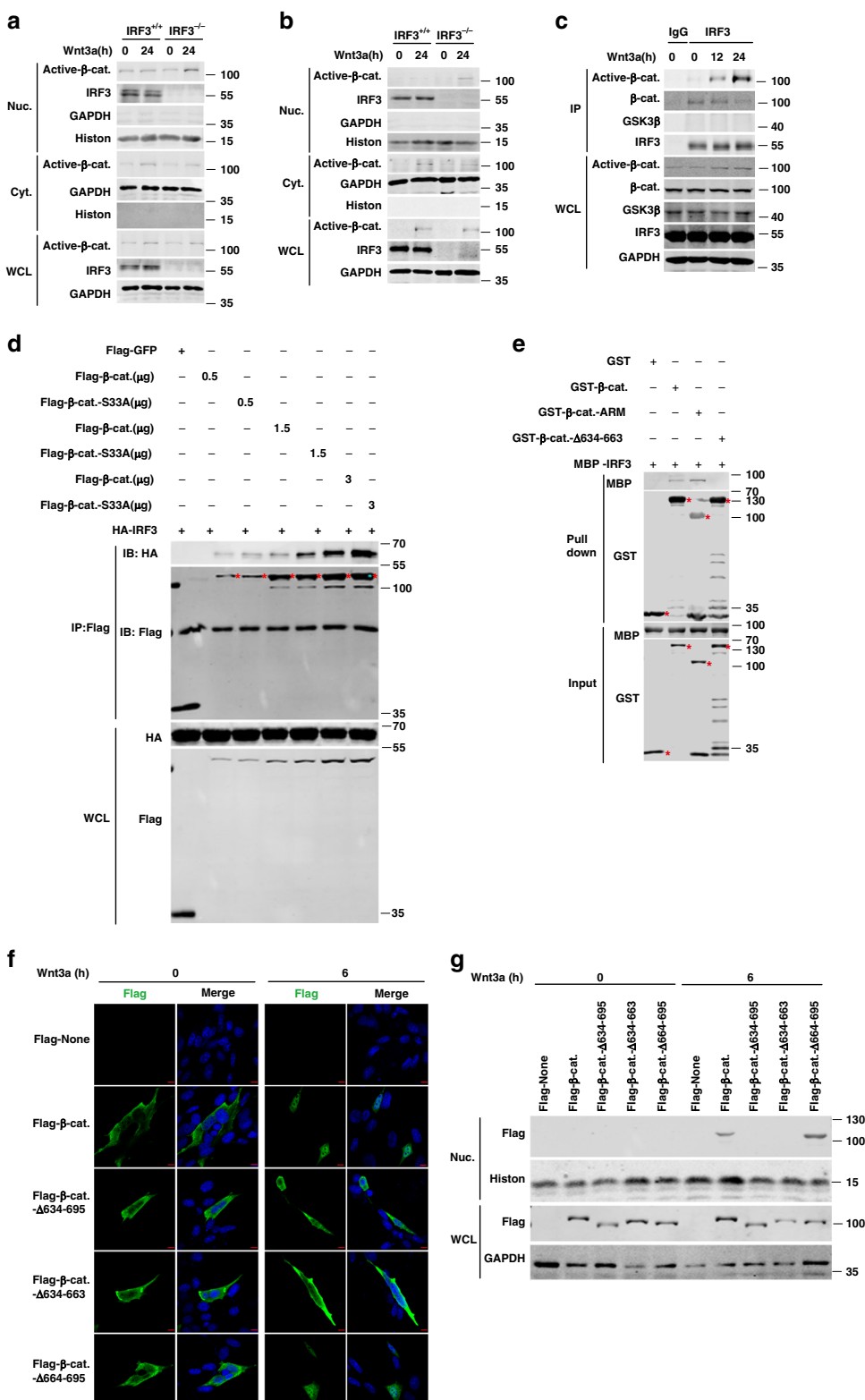

**Fig. 5 IRF3 binds the ARM domain of β-catenin and prevents its nucleus translocation. a**, **b** Nucleocytoplasmic separation and immunoblot analysis of Active-β-catenin in HCT116 (**a**) and H1299 (**b**) cells after treated with wnt3a-conditioned medium. **c** Immunoblot analysis of the endogenous interaction between active-β-catenin, β-catenin, or GSK3β and IRF3 with anti-IRF3 immunoprecipitates in HCT116 cell line extracts after treated with wnt3a-conditioned medium. **d** Immunoblot analysis of the interaction between β-catenin or β-catenin-S33A and IRF3 with anti-FLAG immunoprecipitates in HEK293T cell line. **e** Pull-down analysis the interaction between GST-β-catenin, GST-β-catenin-ARM, or GST-β-catenin-Δ634-663 and MBP-IRF3. **f**, **g** Immunofluorescence (**f**) and nucleocytoplasmic separation (**g**) analysis for the cellular localization of β-catenin or its mutants in HEK293 cell line upon wnt3a-conditioned medium treatment. Red scale bars, 10 μm. Data represent three independent experiments (**a**–**g**). Source data are provided as a Source data file. See also Supplementary Fig. S5.

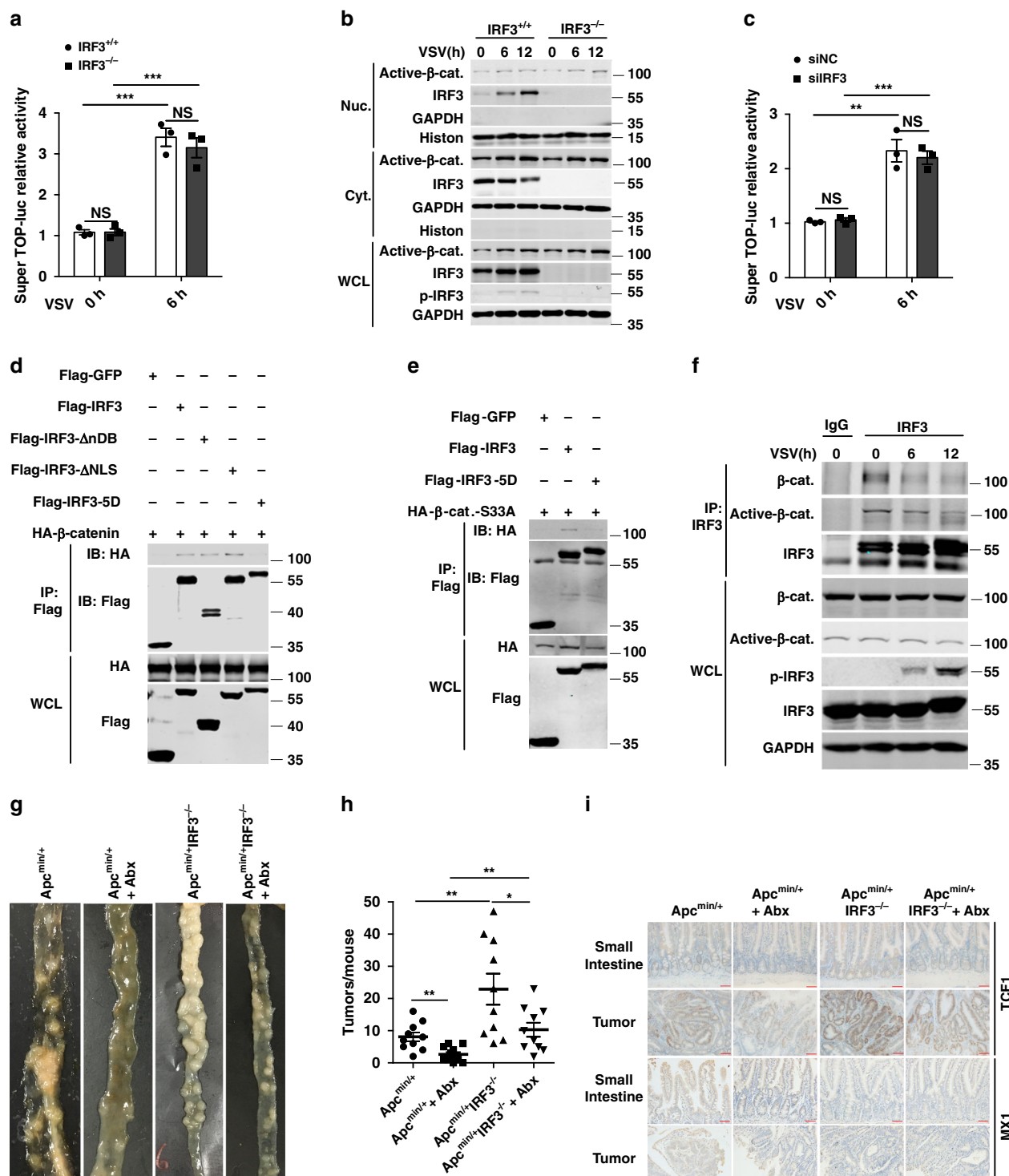

**Fig. 6 Activation of IRF3 by PRR signaling relieves its inhibition on Wnt signaling. a** TOPflash-relative luciferase activity analysis for VSV treatment in IRF3-knockout HCT116 cells. **b** Nucleocytoplasmic separation and immunoblot analysis of active-β-catenin (active-β-cat.) and IRF3 activation in HCT116 cas9 cells after treated with VSV. **c** TOPflash-relative luciferase activity analysis for VSV treatment in siNC and siIRF3 HCT116 cells. **d** Immunoblot analysis for the interaction between IRF3, IRF3-ΔnDB, IRF3-ΔNLS, or IRF3-5D and β-catenin with anti-FLAG immunoprecipitates of HEK293T cells. **e** Immunoblot analysis for the interaction between IRF3 or IRF3-5D and β-catenin-S33A with anti-FLAG immunoprecipitates in HEK293T cells. **f** Immunoblot analysis for the endogenous interaction between active-β-catenin or β-catenin and IRF3 with anti-IRF3 immunoprecipitates in HCT116 cell line extracts treated with VSV. **g** Representative images of the small intestine tumors from 5-month-old Apc$^{min/+}$ and Apc$^{min/+}$ IRF3$^{-/-}$ mice with 4 months Abx treatment. **h** The small intestine tumor counts from Apc$^{min/+}$ and Apc$^{min/+}$ IRF3$^{-/-}$ mice with Abx treatment ($n = 10$, $n = 11$, $n = 10$, $n = 10$). **i** Standarized TCF1 and MX1 immunostaining of the small intestine and tumors from Apc$^{min/+}$ and Apc$^{min/+}$ IRF3$^{-/-}$ mice with Abx treatment or without Abx treatment. Scale bars, 50 μm. Each symbol represents one mouse (**h**). *$P < 0.05$; **$P < 0.01$; ***$P < 0.001$; NS not statistically significant by two-tailed $t$ test (**a**, **c**, **h**). Data are from two (**g–i**) or three (**a–f**) independent experiments and are presented as mean ± s.e.m. in **a**, **c**, **h**. Source data are provided as a Source data file. See also Supplementary Fig. S6.

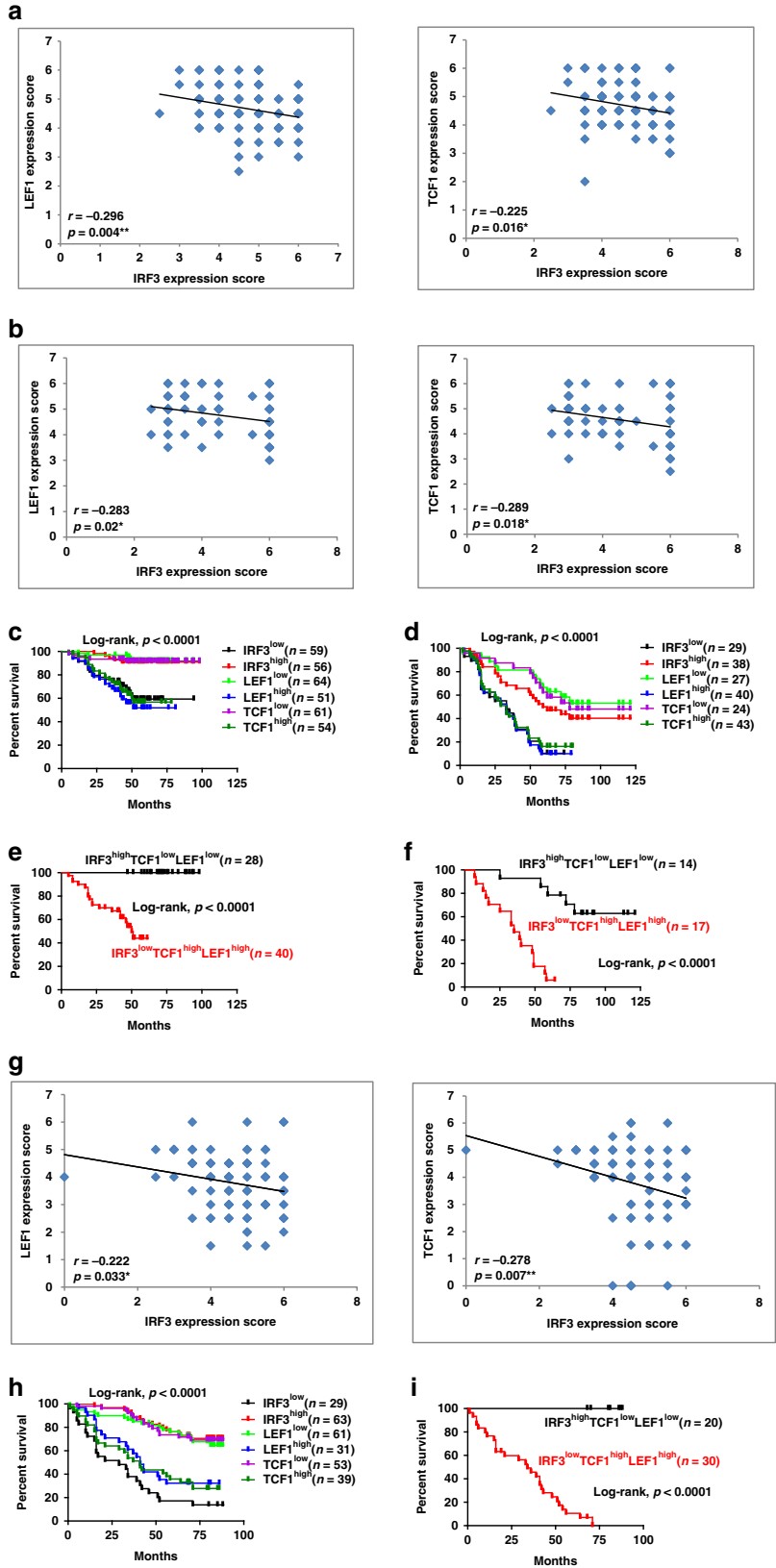

The innate immune system represents an immediate response to defend the host against external agents in organs[45]. For instances, interactions with microorganisms, such as influenza viral from air typically leads to phosphorylating IRF3 and NF-κB, resulting in the production of type I IFN, pro-inflammatory cytokines, and chemokines in lung[46]. The liver is constantly exposed to food, hepatitis B/C virus (H B/C V), and microbial products from the intestine via the portal venous blood[47]. In human lung adenocarcinoma and hepatocellular carcinoma patients, we also found the level of IRF3 had a strong inverse

**Fig. 7 IRF3 expression correlates with the activation of Wnt signaling and the survival of CRC, lung adenocarcinoma, and hepatocellular carcinoma patients. a, b** Correlation analysis for IRF3, LEF1, and TCF1 expression in CRC patients ($n = 115$) (**a**) or in human lung carcinomas ($n = 67$) (**b**). Fisher's exact test. **c, d** Kaplan–Meier analysis for overall survival in a set of CRC patients (**c**) or human lung carcinomas (**d**) according to IRF3, LEF1, and TCF1 expression. **e, f** Combined expression status of IRF3, LEF1, and TCF1 in a set of CRC patients (**e**) or human lung carcinomas (**f**). Log-rank test, log rank, $p < 0.0001$. **g** Correlation between IRF3 expression and LEF1 or TCF1 expression in human hepatocellular carcinoma patients. $n = 92$ cases, Fisher's exact test. **h, i** Kaplan–Meier analysis for the overall survival in a set of hepatocellular carcinoma patients according to IRF3, LEF, and TCF1 expression (**h**), or combined expression status of IRF3, LEF1, and TCF1 (**i**). *$P < 0.05$; **$P < 0.01$ by two-sided Pearson correlation coefficient (**a, b, g**). Log-rank test, log rank, $P < 0.0001$. *$P < 0.05$; **$P < 0.01$; ***$P < 0.001$ (**c–f, h, i**). See also Supplementary Fig. S7.

relation to the expression of TCF1 and LEF1. Given that Wnt signaling plays a critical role in the development of lung adeno-carcinoma[28] and hepatocellular carcinoma[48], whether and how IRF3 activation is involved in lung adenocarcinoma and HCV-induced hepatocellular carcinoma are worthy to be explored.

Jiao et al. demonstrated that IRF3 promotes *Helicobacter pylori* and MNNG-induced gastric tumor formation via promoting YAP activation[25]. Our findings, however, provided strong evidence that IRF3 served as an inhibitor of CRC via inhibiting Wnt signaling. Furthermore, we did not observe any change of YAP activation in the absence of IRF3 in the colon and tumor tissue from mice upon AOM/DSS treatment (Supplementary Fig. S3k, l). Meanwhile, downregulation or overexpression of IRF3 in human gastric carcinoma cell lines BGC-823 (Supplementary Fig. S8a–c) and HGC-27 (Supplementary Fig. S8d–f) did not affect the Wnt target or associated genes expression upon wnt3a treatment. These results indicated that IRF3 affects different signaling pathway in different cells. In addition, in the *H. pylori* and MNNG-induced GC mice model, chronic *H. pylori* infection induces chronic gastritis, precancerous lesions, metaplasia, dysplasia, and gastric cancer, and MNNG is an activated *N*-nitroso compound, which causes chromosomal aberration, point mutation, and DNA damage[49]. While AOM/DSS-induced CRC is based on the chemical alkylation of DNA to facilitate base mis-pairings through AOM and chronic colonic inflammation triggered by administration of the irritant DSS[50]. These might also explain the different roles of IRF3 in these two models.

In conclusion, we described here a noncanonical function of IRF3, which is active in its resting state to inhibit the nuclear import of β-catenin. This unexpected regulation thus links the gut microbiota to the proliferation of intestinal epithelium and the development of CRC, via the IRF3-β-catenin axis identified here.

## Methods

**Mice.** IRF3 and IFNA1R-deficient mice were kindly provided by Pr. Charles. A. Hales (Harvard Medical School. Both of mice and their littermates with a C57BL/6 background were used in this study. The progeny of IRF3$^{+/-}$ intercrosses were genotyped by PCR analysis of DNA isolated from the tail using the following three primers: 5′-GAACCTCGGGAGTTATCCCGAAGG-3′, 5′-GTTTGAGTTATCCCTGCACTTGGG-3′, 5′-TCGTGCTTTACGCTATCGCCGCTCCCGATT-3′. The progeny of IFNAR$^{+/-}$ intercrosses were genotyped by PCR analysis of DNA isolated from the tail using the following three primers: 5′-CGAGGCGAAGTGGTTAAAAG-3′, 5′-ACGGATCAACCTCATTCCAC-3′, 5′-ATTCGCCAATGACAAGACG-3′. Apc$^{min/+}$ mice were purchased from The Jackson Laboratory. IRF3$^{fl/fl}$ (loxP knock-in) mice were generated using CRISPR/Cas9 in C57BL/6 mice, which finished by Institute of Laboratory Animal Sciences, Chinese Academy of Medical Sciences. PCR genotyping of tail DNA was used the primers F1 (5′-GAAATAGTGGGAAAGTATGAGAACG-3′), F2 (5′-CCGCAACACTTCTTTCCG-3′), F3 (5′-GTCCAGAGCTGCACACACATTGT-3′), and F4 (5′-TCCCTGTGCCTCTGAGATTC-3′). The mice were genotyped using primers F1/F2 (wild type-244 bp and mutant-332 bp) and F3/F4 (wild type-612 bp and mutant-706 bp), giving rise to two bands (mutant-332 bp and mutant-706 bp) in homozygous IRF3$^{fl/fl}$ animals, two bands (wild type-244 bp and mutant-332 bp or wild type-612 bp and mutant-706 bp) in heterozygous IRF3$^{fl/+}$ animals and two bands (wild type-244 bp and wild type-612 bp) in wild-type animals. Villin$^{cre}$ mice were purchased from The Jackson Laboratory. All the mice were kept in specific pathogen-free conditions. The housing ambient temperature for the mice is between 20 and 25 °C, the humidity is 60%, and 12 h dark/12 h light cycle. All animal experiments were performed in accordance with protocols approved by the Scientific Investigation Board of Zhejiang University. The animal experiments were performed with

approval from the Institutional Animal Care and Scientific Investigation Board of Zhejiang University.

**AOM/DSS model of colorectal tumorigenesis.** Male and female mice were used at the age of 6 weeks, and then were injected intraperitoneally with 10 mg of AOM (A5486, Sigma) per kg body weight. Five days later, 2.5% DSS (MP Biologicals) was given in the drinking water for 5 days followed by regular drinking water for 2 weeks. This cycle was repeated twice with 2.5% DSS, and mice were sacrificed on day 90. For day 15 samples, mice were injected with AOM, and after 5 days, they were fed with 2.5% DSS for 5 days. Mice were then fed with regular water for 5 days and sacrificed. According to the diameter of the tumors in mice colon on day 90 of AOM/DSS model, we divided them into three group: small tumors, <1 mm; medium tumors, 1 mm ≤ and ≤ 2 mm; large tumors, >2 mm. Tumor load was calculated according to the following formula: tumor load = (number of small tumors) × 1 + (number of medium tumors) × 2 + (number of large tumors) × 3.

**Colonic MRI.** Colonic MRI was performed as reported before[51]. All the mice were placed in the supine position at the center of the mouse coil. The mice were anesthetized by intraperitoneal injection of 4% chloral hydrate (400 mg/kg). A cleansing enema with water was administered 20 min after the liquid enema (Gd-FITC-SLNs), and imaging session was subsequently undertaken after distending the colorectum by 1 mL of room air through a 1-mL syringe and a 24-gauge cannula (Xindeyi Medical Instrument Co. Ltd., Hangzhou, China). Leakage from the rectum was prevented through a small rubber seal placed into the anus of each mouse.

**RNA-seq analysis.** The library construction and sequencing was performed at Shanghai Biotechnology Corporation, and data were also analyzed by Shanghai Biotechnology Corporation. But we performed further analysis of the sequencing results as follows: first, we screen the upregulated genes in IRF3$^{+/+}$ tumor compared with IRF3$^{+/+}$ colon, which is named "WT group", and get "KO group" in the same way. Second, we use the keyword "proliferation" to pick up the upregulated genes in "WT group" and "KO group", which are named "WT" and "KO", respectively. Third, we get 92 genes that upregulated both in "WT" and "KO", 16 genes that only upregulated in "WT" and 65 genes that only upregulated in "KO" (Supplementary Fig. S3a), and obtained the related signal pathway through PANTHER database (http://www.pantherdb.org/) by the 65 genes. The 65 genes are listed in Supplementary Table 2.

**Intestine organoid culture.** Mouse colon stem cells were cultured using Intesti-Cult organoid growth medium according to the manufacturer's instructions (06005, STEMCELL Technologies). The whole colon was removed from untreated IRF3$^{+/+}$ and IRF3$^{-/-}$ mice, and rinsed with ice cold PBS. Repeat this process until the supernatant no longer contains any visible debris. The colon was cut into 5 mm pieces and placed into ice cold 5 mM EDTA-PBS. Colon segments were incubated in Gentle Cell Dissociation Reagent (07174, STEMCELL Technologies), rotated at $350 \times g$ for 15 min at room temperature, followed by resuspension in PBS supplemented with 0.1% BSA (A6003, Sigma). Dissociated colon crypts were filtered through 70 mm strainers. Dissociated colon crypts were resuspended in Dulbecco's modified Eagle's medium (DMEM)/F12 medium with 15 mM HEPES (36254, STEMCELL Technologies), counted, and resuspended in Intesticult organoid growth medium (with wnt3a conditional medium) and Matrigel (356230, Corning) in a 1:1 ratio. Cells were plated in 24-well culture plates (3738, Corning).

**Cell culture, plasmid transfection, and siRNA silencing.** HCT116, SW620, BGC-823, and HGC-27 cell lines were obtained from American Type Culture Collection. H1299 and L-wnt3a cell lines were obtained from Professor Ping Wang in School of Medicine and School of Life Science and Technology, Tongji University. HEK293 and HEK293T cells were got from Dr. Huazhang An, Second Military Medical University, Shanghai, and grown in DMEM supplemented with 10% fetal bovine serum (Gibico). Scramble siRNA and IRF3-targeted siRNA were transfected in HCT116 and H1299 cells, using INTERFERin@ according to the manufacturer's protocol. The following siRNA oligonucleotide sequences were used: IRF3 siRNA (5′- AGACAUUCUGGAUGAGUUA-3′).

**Ethics**. The experimental license to use human paraffin-embedded colon sections was approved by the Medical Research Ethics Committee of Zhejiang University. In addition, informed consent was obtained from all of the subjects involved, and the experiments were conducted according to the principles expressed in the Declaration of Helsinki.

**Statistical analysis**. Statistical specifications of each experiment such as number of animals, number of tumors, biological replicates, technical replicates, precision measures (mean and ±s.e.m.), and the statistical tests used are provided in the figures and figure legends. Unpaired Student's $t$ test was used to calculate the $P$ values for comparisons of tumor numbers, tumor load, relative mRNA expression levels, or quantitative evaluation of immunohistochemical staining. Correlation studies of immunohistochemically stained CRC, lung adenocarcinoma, and hepatocellular carcinoma tissue samples were analyzed using the Pearson correlation factor $r$. Kaplan–Meier survival analysis was performed using the software Prism v5.0 (Graphpad Software) with the log-rank (Mantel–Cox) test.

**Reporting summary**. Further information on research design is available in the Nature Research Reporting Summary linked to this article.

## Data availability

The RNA-seq data have been deposited in the GEO database under the accession code GSE155777. The differential genes from RNA-seq data are analyzed with PANTHER database (http://www.pantherdb.org/). All the other data supporting the findings of this study are available within the article and its supplementary information files, and from the corresponding author upon reasonable request. A reporting summary for this article is available as a Supplementary Information file. Source data are provided with this paper.

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

## Acknowledgements

We thank Dr. Ping Wang and Dr. Huazhang An for the helpful discussion. We also thank Dr. Xin Chen for the helpful advice on RNA-seq results analysis. We thank Key Laboratory of Immunity and Inflammatory Diseases of Zhejiang Province. This work was supported by grants from the National Natural Science Foundation of China (31970899, 82071774 and 81571738), Key Research and Development Program of Zhejiang Province (2019C03014).

## Author contributions

X.W. and M.T. designed experiments and analyzed data. M.T. and X.W. conducted biochemical and cellular studies. M.T., X.W., and W.L. performed animal studies. M.T. and L.C. performed in vivo MRI analyses. J.S. and X.C. provided clinical samples. X.W. and W.L. provided the help for pathological analysis. M.T. provided the help for IRF3 mRNA expression analysis from TCGA database. X.M.W., L.S., D.L., and X.C provided reagent. X.W., M.T., and P.X. wrote and revised the manuscript, which was edited by all authors.

## Competing interests

The authors declare no competing interests.
