## [Peer Review File · Nature Communications]

Reviewers' comments:

Reviewer #1 (Remarks to the Author):

In this manuscript Tian and colleagues show that loss of irf3 in the epithelium ncreases tumourigenesis following Aom/DSS and in the apc min mouse. This data is clear and well done. The postulated mechanism is via increased b-catenin signalling. Whilst it is clear this b-catenin goes up ,I feel there are extra experiments needed here. Finally I was not convinced with the antibiotic experiments.

Specific points

- 1) Wnt signalling is known to be active in the stem cells of normal crypts. Can the authors explain why irf3 only plays a roles in the cancer models and not in normal homeostasis? There are no changes in wnt targets. Could the authors do rnascope for stem cell markers and wnt targets in normal crypts (si and colon). I note there are differences in organoids.
- 2) the premise of this manuscript is that increasing wnt - b-catenin signalling will exacerbate tumorigenesis. There has been a lot literature about the "just right " wnt signalling hypothesis and this is not necessarily the case. In the min mouse tumour location proximal-distal in the small intestine alters (see studies from leedham, Tomlinsom, Bienz and sansom). Tumour location need to presented and previous data on just right signalling addressed and discussed carefully.
- 3) Following on from this icg001 is relatively controversial over how it inhibits b-catenin/cbp. Could the authors use other approaches to inhibit wnt eg tanks, porc? Ideally genetic reduction of b-catenin to heterozygosity would have been optimal.
- 4) I was not convinced about the preferential impact of antibiotics on the min irf3 v min. A proper statistical comparison needs to be done comparing if the reduction of polyps is more.
- 5) expression of irf3 in the different consensus subtype of crc is needed to see if they follow a specific different distribution
- 6) in the hcc samples do the authors know whether b-catenin is mutated and/or rspo amplification.
- 7) I'm not really sure that the curves showing lef low, Tcf low irf3 low really help. These are very different crc. For this paper the importance is irf3 levels in tcf lef high situations. I would not over interpret these correlations

Reviewer #2 (Remarks to the Author):

In this study entitled "IRF3 prevents colorectal tumorigenesis via inhibiting the nuclear translocation of β -catenin" by Miao Tian et al., the authors identified IRF3 as a potential tumor suppressor. They demonstrated that IRF3 deficient mice are hyper-susceptible to the development of intestinal tumor in AOM/DSS and Apcmin/+ models. Knockout of of IRF3 promotes proliferation of intestinal epithelial cells. From mechanism studies, they demonstrated that IRF3 suppresses the CRC via inhibiting Wnt signaling by an RNA-sequencing approach. Furthermore, they demonstrated that IRF3 bound with and prevented β -catenin nucleus translocation. Overall, the studies are well-performed and the conclusion regarding the deficiency of IRF3 promotes proliferation of intestinal epithelial cells and enhances colorectal tumorigenesis are largely supported by the data. The data are of a very good technical quality and their findings have a significant impact on the field of cancer research. However, the data to support that IRF3 prevents β -catenin nucleus translocation is weak. There is no data to confirm β -catenin nucleus translocation in IRF3 deficient tumor samples.

Specific issues:

1. Although the β -catenin nucleus translocation is slightly increased in IRF3 deficient cells as showing in Fig. 5 A and B, if the data is compared with the data in Fig. S5E, Wnt3a-mediated β -catenin nucleus translocation in IRF3 deficient cells will be decreased compared with Flag-none cells. Nuclear IRF3 in Fig 5 should be replace with a clean blot.

2. The constitutively active form of IRF3 (IRF3-5D) fails to bind β -catenin could be due to the changes in amino acids, subcellular localization or dimerization. It would be interesting to see if IRF3 NES (constitutive nuclear localization) or IRF3 5A (inactive form of IRF3) binds and prevents β -catenin nucleus translocation.
3. The authors show that virus infection induces expression Wnt target genes in wild type cells but not in IRF3 deficient cells. The data also shows the same basal levels of these Wnt target genes is expressed in both cells (Fig. S6). If IRF3 negatively regulates Wnt signaling, why knockout IRF3 does not enhance Wnt signaling? Are authors proposing that IRF3 is required for virus-mediated activation of Wnt signaling? If virus-mediated activation of Wnt signaling is due to the relief the IRF3 inhibition, IRF3 deficient cells should constitutively activate the Wnt signaling.
4. Jiao et al have demonstrated that IRF3 knockout inhibited Helicobacter pylori and MNNG induced gastric tumor formation in C57BL/6 mouse model, suggesting a role of IRF3 in tumorigenesis (ref. 25). The authors should discuss possible mechanisms involved in these opposite functions of IRF3 in tumorigenesis.
5. The authors refer the mimics constitutive activate of IRF3 (IRF3-5D) to reference 20. However, in reference 20, only IRF3 5SD is described. Is the IRF3-5D used in this manuscript the same as the one first described in Lin et al 1998 MCB (replaced 4 serine and 1 threonine residues with aspartic acid)? If not, the 5 serine residues should be indicated.
6. Does IRF3 inhibit Wnt signaling in GC cells HGC-27 and BGC-823?
7. What is wnt3a-conditioned medium?
8. The abbreviations should be defined when they first time used in the text.
9. Page 3, the tumor p53 should be replaced with the tumor protein p53.

To Reviewer 1:

Thanks very much for your comments and suggestions regarding our manuscript entitled “IRF3 prevents colorectal tumorigenesis via inhibiting the nuclear translocation of β -catenin” (NCOMMS-19-35939). According to your suggestions, we have added some new data and re-organized the manuscript. If you have any further questions, please inform us and we can discuss them further.

Specific points

1. **Question:** *Wnt signalling is known to be active in the stem cells of normal crypts. Can the authors explain why irf3 only plays a role in the cancer models and not in normal homeostasis? There are no changes in wnt targets. Could the authors do rna-scope for stem cell markers and wnt targets in normal crypts (si and colon). I note there are differences in organoids.*

Response: IRF3 deficiency promotes Wnt target gene expression in colon from AOM-DSS treated mice but not in the mice without AOM/DSS treatment (**Fig.3C-D, S3B-G**). Consistently, IRF3 does not affect the basal level of Wnt signalling in cancer cell (**Fig.4K, S4C-E, S4G-J, S4R-U, Fig.5A-B, S5A-E, S5O-P**). Lots of molecules are demonstrated to regulate Wnt signalling pathway, which converge at regulation of β -catenin protein stability. In resting state, β -catenin is phosphorylated and degraded by the proteasome to keep very low level of β -catenin in cytoplasm to maintain homeostasis. When the pathway is activated, for example upon wnt3a stimulation, β -catenin is stabilized, accumulated and activated in the cytoplasm, which leads to its nuclear translocation and induction of Wnt target genes.

In our study, we show that IRF3 directly binds to the active form of β -catenin and inhibits its nuclear translocation, which could be a second level of regulation of β -catenin function and Wnt signaling. IRF3 knockout might be not sufficient to cause the notable change in basal level of Wnt signaling. When the Wnt signaling is activated, β -catenin is accumulated in the cytoplasm and trans-locates into the nucleus. In that case, IRF3 binds the active- β -catenin and inhibits the Wnt signaling. Maybe that is reason why IRF3 deficiency only affects the Wnt signaling in the cancer models but not in normal

homeostasis.

According to your suggestion, we performed RNA in situ hybridization (ISH) for stem cell marker (Lgr5) and Wnt target gene (Axin2) in normal crypts (SI and colon) from IRF3^{fl/fl} and IRF3^{fl/fl}Villin^{cre} mice. As shown in the **Fig. S3H** in the current manuscript, the RNA level of Lgr5 and Axin2 in normal crypts showed no difference in IRF3^{fl/fl} and IRF3^{fl/fl}Villin^{cre} mice, indicating that IRF3 deficiency in intestinal epithelium does not affect Wnt signaling in normal homeostasis, but promotes the Wnt signaling in the AOM/DSS model.

2. Question: *The premise of this manuscript is that increasing wnt - b-catenin signalling will exacerbate tumorigenesis. There has been a lot literature about the “just right” wnt signalling hypothesis and this is not necessarily the case. In the min mouse tumor location proximal-distal in the small intestine alters (see studies from leedham, Tomlinsom, Bienz and sansom). Tumour location need to be presented and previous data on just right signalling addressed and discussed carefully.*

Response: Polyp distribution was markedly different between Apc^{1322T} and Apc^{min/+} strain: Apc^{1322T} animals develop a heavy tumour burden in the proximal small bowel (SB1) (segments 1 and 2) whereas Apc^{min/+} mice have relatively more polyps in the distal small bowel (SB3) (segments 2 and 3) [1]. In our Apc^{min/+} mice model, the tumour were mainly located in SB1 and SB3. The tumours located in the whole small intestine were summed in **Fig.1** and **Fig.6**. We have added this description on **page 6** in the revised manuscript.

According to the ‘just-right’ theory, an optimal but not excessive level of accumulation of nuclear β -catenin is considered favourable for tumorigenesis. As shown in **Fig.1**, IRF3 deficiency promoted the tumorigenesis both in AOM/DSS and Apc^{min/+} mouse model, indicating the IRF3-promoting β -catenin activation was in the optimal range of Wnt activation. We have added this description on **page 14** in the revised manuscript.

3. Question: *Following on from this icg001 is relatively controversial over how it inhibits b-catenin/cbp. Could the authors use other approaches to inhibit wnt eg tanks, porc? Ideally genetic reduction of b-catenin to heterozygosity would have been optimal.*

Response: Thanks for your suggestions. We applied another inhibitor G007-LK, which was a selective inhibitor of TNKS1 and TNKS2 in AOM/DSS model. As shown in **Fig. S3N-Q** in the current manuscript, G007-LK treatment abolished the increase of tumorigenesis (**Fig. S3N-P**) and Wnt signaling activation (**Fig. S3Q**) in IRF3^{fl/fl}Villin^{cre} mice. These results indicate IRF3 restrains colon tumorigenesis via inhibiting Wnt/ β -catenin pathway.

Actually, we have crossed IRF3^{fl/fl}Villin^{cre} mice with Ctnnb1^{fl/fl} (β -catenin^{fl/fl}) mice for almost one and a half years. Unfortunately we failed to obtain the Ctnnb1^{fl/fl}IRF3^{fl/fl}Villin^{cre} and Ctnnb1^{fl/fl}Villin^{cre} mice. It has been reported that intestinal stem cells were induced to terminally differentiate upon deletion of β -catenin, resulting in a complete block of intestinal homeostasis and fatal loss of intestinal function [2]. Therefore, we speculate the Ctnnb^{fl/fl}Villin^{cre} strain might be embryonic lethal.

4. Question: *I was not convinced about the preferential impact of antibiotics on the min irf3 v min. A proper statistical comparison needs to be done comparing if the reduction of polyps is more.*

Response: We had performed two independent experiments that the Apc^{min/+} and Apc^{min/+}IRF3^{-/-} mice were treated with or without Abx. The data from another independent experiment were shown as below: the decreased rate of tumor number in Apc^{min/+} mice (70.4%) was higher than those in Apc^{min/+}IRF3^{-/-} mice (55.7%): Abx treatment reduced the average number of tumors per mice from 8.8 to 2.6 in Apc^{min/+} mice while reduced the number of tumors per mice from 35.0 to 15.5 in the Apc^{min/+}IRF3^{-/-} mice. As shown in **Table S1** in the current manuscript, the reduction of polyps in Apc^{min/+} mice was more than that in Apc^{min/+}IRF3^{-/-} mice with Abx treatment. These results indicate that triggered activation of IRF3 by gut microbiota contributes to the microbiota-induced tumorigenesis.

5. **Question:** *Expression of irf3 in the different consensus subtype of CRC is needed to see if they follow a specific different distribution.*

Response: Thanks for your suggestions. According to the histopathological characteristics in clinicopathological features of the colorectal cancer patients, we collected the clinical samples which were classified as adenocarcinomas-NOS (Not Otherwise Specified). CRC variants (such as mucinous adenocarcinoma, signet ring cell carcinoma, serrated adenocarcinoma and so on) were not included in our study. We have added this description on **page 34-35** in the supplementary experimental procedures in the revised manuscript.

6. **Question:** *In the HCC samples do the authors know whether b-catenin is mutated and/or rspo amplification.*

Response: According to your suggestion, the expression of RSPO3 in HCC samples (n=84) was detected. As shown in the **Response Fig. 1**, the expression of RSPO3, LEF1 and TCF1 in tumor tissue was higher than that in the paracancerous tissue. These results indicated that Wnt signaling was activated during liver tumorigenesis.

Response Figure 1. Increased expression of RSPO3, LEF1 and TCF1 in tumor tissue of HCC samples. (A) Standardized RSPO3 (upper), LEF1 (middle) and TCF1 (down) immunostaining of paracancerous and tumor tissues of HCC patients, scale bar, 50 μ m. **(B)** Quantification of the expression score of RSPO3 (upper), LEF1 (middle) and TCF1 (down) of HCC patients (n=84) paracancerous and tumor tissues. *P< 0.05; **P< 0.01; ***P< 0.001 by two tailed *t-test* **(B)**. Data are presented as mean \pm s.e.m. in **B**.

7. Question: *I'm not really sure that the curves showing lef low, Tcf low irf3 low really help. These are very different CRC. For this paper the importance is irf3 levels in tcf lef high situations. I would not over interpret these correlations.*

Response: Yes, you are right. In our study, we identified that the expression level of IRF3 protein was inversely correlated with levels of TCF1 and LEF1 (**Fig.7A**), that is the low IRF3 levels in TCF or LEF high situation and high IRF3 levels in TCF or LEF low situation. The curves showing LEF^{low} TCF^{low} IRF3^{low} group was not shown in the current manuscript.

To Reviewer 2:

Thanks very much for your comments and suggestions regarding our manuscript entitled “IRF3 prevents colorectal tumorigenesis via inhibiting the nuclear translocation of β -catenin” (NCOMMS-19-35939). According to your suggestions, we have added some new data and re-organized the manuscript. If you have any further questions, please inform us and we can discuss them further.

Specific issues:

*1. **Question:** Although the β -catenin nucleus translocation is slightly increased in IRF3 deficient cells as showing in Fig. 5 A and B, if the data is compared with the data in Fig. S5E, Wnt3a-mediated β -catenin nucleus translocation in IRF3 deficient cells will be decreased compared with Flag-none cells. Nuclear IRF3 in Fig 5 should be replace with a clean blot.*

Response: In the **Fig.5A-B**, the IRF3 knockout (IRF3^{-/-}) HCT116 and HCT1299 cells were constructed using the CRISPR/Cas9 gene-editing system and stimulated with wnt3a. Whereas in the **Fig.S5D-E**, HCT116 and HCT1299 cells transiently transfected with empty vector (flag-none) or IRF3 expressing vector (flag-IRF3) were stimulated with wnt3a. Therefore, these experiments were performed independently. The band intensity was related to the exposures time. Maybe that was the reason why wnt3a-mediated β -catenin nucleus translocation in IRF3 deficient cells in **Fig. 5B (Response Fig. 2A)** was decreased compared with flag-none cells in **Fig. S5E**. Wnt3a-mediated β -catenin nucleus translocation in IRF3 deficient cells is increased (**Fig. 5B/Response Fig. 2A**) if compared with flag-none cells in the **Response Fig. 2B** which is from another independent experiment. We have re-run the sample and the new band of nuclear IRF3 in **Fig. 5A** was provided in the current manuscript.

Response Figure 2. IRF3 inhibits active-β-catenin nucleus translocation. (A) The picture in Fig. 5B. Nucleocytoplasmic separation and immunoblot analysis of Active-β-catenin in IRF3^{+/+} or IRF3^{-/-} H1299 cells after treated with wnt3a-conditioned medium. **(B)** Nucleocytoplasmic separation and immunoblot analysis of active-β-catenin in IRF3 overexpressed H1299 cells with wnt3a-conditioned medium treatment.

2. **Question:** *The constitutively active form of IRF3 (IRF3-5D) fails to bind β-catenin could be due to the changes in amino acids, subcellular localization or dimerization. It would be interesting to see if IRF3 NES (constitutive nuclear localization) or IRF3 5A (inactive form of IRF3) binds and prevents β-catenin nucleus translocation.*

Response: Yes, you are right. The constitutively active form of IRF3 (IRF3-5D) fails to bind β-catenin could be due to the changes in amino acids, subcellular localization or dimerization. According to your suggestion, we constructed IRF3 5A (inactive form of IRF3) and performed CO-IP in HEK293 with HA-β-catenin. As shown in the Response Fig. 3A, HA-β-catenin is co-immunoprecipitated with Flag-IRF3-5A in HEK293T cells. Ectopic expression of Flag-IRF3-5A inhibited the nuclear translocation of HA-β-catenin in HEK293T cells (Response Fig. 3B). Meanwhile, ectopic expression of IRF3-5A in HCT116 and H1299 cells restrained the nuclear translocation of active-β-catenin (Response Fig. 3C-D). These results indicate the cytoplasmic IRF3 in resting state binds and prevents β-catenin nucleus translocation.

Response Figure 3. IRF3-5A binds and prevents β -catenin nucleus translocation. (A) Immunoblot analysis for the interaction between IRF3-5A and β -catenin with anti-Flag immunoprecipitates in HEK293T cells. **(B)** Nucleocytoplasmic separation and immunoblot analysis of HA- β -catenin in HA- β -catenin and flag-none or flag-IRF3-5A co-overexpressed HEK293 cells with wnt3a-conditioned medium treatment. **(C-D)** Nucleocytoplasmic separation and immunoblot analysis of active- β -catenin in IRF3 overexpressed HCT116 **(C)** and H1299 **(D)** cells with wnt3a-conditioned medium treatment.

3. **Question:** *The authors show that virus infection induces expression Wnt target genes in wild type cells but not in IRF3 deficient cells. The data also shows the same basal levels of these Wnt target genes is expressed in both cells (Fig. S6). If IRF3 negatively regulates Wnt signaling, why knockout IRF3 does not enhance Wnt signaling? Are authors proposing that IRF3 is required for virus-mediated activation of Wnt signaling? If virus-mediated activation of Wnt signaling is due to the relief the IRF3 inhibition, IRF3 deficient cells should constitutively activate the Wnt signaling.*

Response:

We thank the reviewer for the reasoning.

We were also mystified by these data. We went back to our cDNA samples (previous Fig.S6A) and performed QPCR assay to check the induction level of IL6 and TNF α . Unexpectedly, there wasn't any increasement of IL6 and TNF α after VSV stimulation in IRF3^{-/-} cells (Response Fig. 4A). More intriguingly, reconstruction with IRF3 in these IRF3^{-/-} cells failed to rescue the expression of IL6 or TNF α with VSV treatment (Response Fig. 4B) and Wnt target genes (Response Fig. 4C). These results suggested to us there could be a problem with our IRF3^{-/-} KO cell line (probably due to off-target effect when we generated our IRF3^{-/-}).

We therefore re-performed the VSV infection experiment in the IRF3^{-/-} HCT116 cas9 strains used in Fig. 4 and Fig. S4. As shown in the in Fig. 6A-B, S6A, VSV infection activated the Wnt pathway in HCT116 cells, as evidenced by luciferase report assay, nucleocytoplasmic separation assay and QPCR assay. Intriguingly, Wnt activation by VSV infection was not affected by IRF3 knockout or knockdown (Fig. 6A-C, S6A-B), but was greatly inhibited by treatment using cycloheximide (CHX, new protein synthesis inhibitor), U0126 (MAPK signaling inhibitor) or BAY 11-7082 (NF- κ B signaling inhibitor) (Fig. S6C). In line with the previous report, IRF3 deficiency significantly inhibited Type I interferon and ISG expression, but had no effect on IL6 and TNF α expression (Fig. S6D-E). Taken together, these results indicated that virus-mediated activation of Wnt signaling is probably due to two mechanisms: On one hand, virus-induced IRF3 activation relieves its inhibition on Wnt signaling. On the other hand, virus infection induces an unknown protein via MAPK and NF- κ B signaling, which in

turn mediates Wnt activation.

Response Figure 4. Characterization of the IRF3 KO cell line used in our previous Fig.S6A. (A-B) Real-time qPCR analysis for IL6 and TNF α in IRF3^{-/-} HCT116 cas9 cells (A) and in rescued HCT116 cas9 cells stimulated with VSV stimulation for indicated time (B). (C) Real-time qPCR analysis for the Wnt target genes in rescued HCT116 cas9 cells stimulated with VSV. *P< 0.05; **P< 0.01; ***P< 0.001; NS, not statistically significant by two tailed t-test (A-C). Data represent three independent experiments (A-C) and are presented as mean \pm s.e.m. in A-C.

4. **Question:** *Jiao et al have demonstrated that IRF3 knockout inhibited Helicobacter pylori and MNNG induced gastric tumor formation in C57BL/6 mouse model, suggesting a role of IRF3 in tumorigenesis (ref. 25). The authors should discuss possible mechanisms involved in these opposite functions of IRF3 in tumorigenesis.*

Response: Jiao et al demonstrated that IRF3 promotes Helicobacter pylori and MNNG induced gastric tumor formation via promoting YAP activation. Our findings, however, provided strong evidence that IRF3 served as an inhibitor of CRC via inhibiting Wnt signaling. Furthermore, we did not observe any change of YAP activation in the absence of IRF3 in the colon and tumor tissue from mice upon AOM/DSS treatment (Fig. S3K-L). Meanwhile, down-regulation or overexpression of IRF3 in human gastric carcinoma cell lines BGC-823 (Fig. S8A-C) and HGC-27 (Fig. S8D-F) did not affect the Wnt target or associated genes expression upon wnt3a treatment. These results indicated that IRF3 affects different signaling pathway in different cells. In addition, in the Helicobacter pylori and MNNG-induced GC mice model, chronic Helicobacter pylori infection induces chronic gastritis, precancerous lesions, metaplasia, dysplasia and gastric cancer, and MNNG is an activated N-nitroso compound, which causes chromosomal aberration, point mutation and DNA damage. While AOM/DSS induced CRC is based on the chemical alkylation of DNA to facilitate base mispairings through AOM and chronic colonic inflammation triggered by administration of the irritant DSS. These might also explain the different role of IRF3 in these two models. We have added the discussion on page 15-16 in the current manuscript.

5. **Question:** *The authors refer the mimics constitutive activate of IRF3 (IRF3-5D) to reference 20. However, in reference 20, only IRF3 5SD is described. Is the IRF3-5D used in this manuscript the same as the one first described in Lin et al 1998 MCB (replaced 4 serine and 1 threonine residues with aspartic acid)? If not, the 5 serine residues should be indicated.*

Response: Sorry for the mistake. The IRF3-5D used in this manuscript is the same as the one described in Lin et al 1998 MCB (replaced 4 serines of 396/398/402/405 and 1

threonine of 404 residues with aspartic acid). We have replaced the reference (ref. 31) in the current manuscript.

6. Question: *Does IRF3 inhibit Wnt signaling in GC cells HGC-27 and BGC-823?*

Response: According to your suggestion, we have downregulated and overexpressed IRF3 in human gastric carcinoma cell lines BGC-823 and HGC-27, and performed real-time PCR to detect the expression of Wnt target and associated gene after wnt3a treatment. As shown in **Fig. S8**, down-regulation or overexpression of IRF3 in human gastric carcinoma cell lines BGC-823 did not affect the Wnt target or associated genes expression upon wnt3a-conditioned medium treatment (**Fig. S8A-C**). Similiar results were obtained in HGC-27 cells (**Fig. S8D-F**). These results indicated that IRF3 does not affect Wnt signaling in GC cells.

7. Question: *What is wnt3a-conditioned medium?*

Response: Wnt3a-conditioned medium is prepared following the guideline of ATCC website (<https://www.atcc.org/products/all/CRL-2647.aspx#culturemethod>) : Firstly, split the L-wnt3a cells 1:10 in 10 mL culture medium (without G418) in 10 cm tissue culture dishes and let the cells grow for 4 days (approximately to confluency); secondly, take off the medium to get the first batch of medium. Add 10 mL fresh culture medium and culture for another 3 days to get the second batch of medium. Finally, mix the first batch with second batch of medium (1:1), which is the wnt-3A conditioned medium. We have added this protocol on **page 38** in the current supplementary experimental procedures in the revised manuscript.

8. Question: *The abbreviations should be defined when they first time used in the text.*

Response: Sorry for the mistake. We have defined the abbreviations in the current manuscript.

9. Question: *Page 3, the tumor p53 should be replaced with the tumor protein p53.*

Response: Sorry for the mistake. We have replaced “the tumor p53” with “the tumor

protein p53” in Page 4 in the current manuscript.

References

1. Leedham, S.J., et al., *A basal gradient of Wnt and stem-cell number influences regional tumour distribution in human and mouse intestinal tracts*. Gut, 2013. **62**(1): p. 83-93.
2. Fevr, T., et al., *Wnt/beta-catenin is essential for intestinal homeostasis and maintenance of intestinal stem cells*. Mol Cell Biol, 2007. **27**(21): p. 7551-9.

REVIEWERS' COMMENTS

Reviewer #1 (Remarks to the Author):

The authors have done a very good job answering all my questions. The paper has benefited from the additional data and i think will make an important contribution of the literature.

Reviewer #2 (Remarks to the Author):

In this resubmission, my original criticisms have been very well addressed. I believe that this is an interesting manuscript suitable for publication at this Journal.